# ARC-AGI WITHOUT PRETRAINING

## ABSTRACT

Conventional wisdom in the age of LLMs dictates that solving IQ-test-like visual puzzles from the ARC-AGI-1 benchmark requires capabilities derived from massive pretraining. To counter this, we introduce *CompressARC*, a 76K parameter model without any pretraining that solves 20% of evaluation puzzles by minimizing the description length (MDL) of the target puzzle purely during inference time. The MDL endows CompressARC with extreme generalization abilities typically unheard of in deep learning. To our knowledge, CompressARC is the only deep learning method for ARC-AGI where training happens only on a single sample: the target inference puzzle itself, with the final solution information removed. Moreover, CompressARC does not train on the pre-provided ARC-AGI "training set". Under these extremely data-limited conditions, we do not ordinarily expect any puzzles to be solvable at all. Yet CompressARC still solves a diverse distribution of creative ARC-AGI puzzles, suggesting MDL to be an alternative, highly feasible way to produce intelligence, besides conventional massive pretraining.

## 1 INTRODUCTION

The ARC-AGI-1 benchmark consists of abstract visual reasoning puzzles designed to evaluate a system's ability to rapidly acquire new skills from minimal input data (Chollet, 2019). Recent progress in LLM-based reasoning has shown impressive skill acquisition capabilities, but these systems still rely on massive amounts of pretraining data. In this paper, we explore how little data is truly required to tackle ARC-AGI by introducing *CompressARC*, a solution method derived from the Minimum Description Length (MDL) principle (Rissanen, 1978). CompressARC performs all of its learning at inference time and achieves 20% accuracy on ARC-AGI-1 evaluation puzzles—using only the puzzle being solved as input data.

The key to CompressARC's extreme data efficiency is its formulation as a code-golfing problem: to find the shortest possible self-contained program that outputs the entire ARC-AGI dataset, with any unsolved grids filled arbitrarily. By Occam's razor, the shortest program is expected to contain the "correct" solutions. An overly basic program might store a hard-coded string of the puzzle data (plus arbitrary solutions) for output; but the program will be too long, implying (by Occam's razor) the outputted solutions will be wrong. On the other hand, finding the optimally shortest program would require exhaustively enumerating many candidate programs, which is computationally infeasible (Solomonoff, 1964; Hutter, 2005). CompressARC strikes a new type of balance by overfitting a neural network to the puzzle data to compress the puzzles into weight matrices; these weights can be hard-coded into the program instead of the puzzles themselves, and then used within the program to recover the memorized puzzles, and also generate solutions. With careful counting of the bit length of the hard-coded weights, this technique converts the combinatorial search of finding the best program into a differentiable optimization problem, allowing us to minimize program length in a reasonable amount of time and still generate good solution predictions.

This framing preserves several attractive properties of the original code-golf formulation, all of which are novel when it comes to neural solutions to ARC-AGI:

- **No pretraining:** Since we begin with the target puzzle(s) in hand, no training phase is required.
- **Inference-time learning:** Program length is minimized solely during inference by optimizing network weights with respect to the target puzzle(s).
- **Minimal data requirement:** Following Occam's razor, we assume strong generalization from the shortest program and use only the puzzle(s) themselves—no additional data is loaded into memory.

Despite never using the training set, performing no pretraining, and having only 76K parameters in its network, CompressARC generalizes strongly, solving 20% of evaluation puzzles and 34.75% of training puzzles—performance that would be impossible for traditional deep learning methods under these constraints. CompressARC's strong performance in this setting suggests that bringing code-golf to other data-limited contexts beyond ARC-AGI (e.g., drug discovery, protein design) may help us extract stronger capabilities in those applications as well.

The remainder of this paper introduces the ARC-AGI benchmark (Section 2), details the problem framing (Section 3), describes CompressARC's architecture (Section 4), presents empirical results (Section 5), and concludes with a discussion of implications (Section 6).

## 2 BACKGROUND: THE ARC-AGI BENCHMARK

ARC-AGI-1 is an artificial intelligence benchmark designed to test a system's ability to acquire new skills from minimal examples (Chollet, 2019). Each puzzle in the benchmark consists of a different hidden rule, which the system must apply to an input colored grid to produce a ground truth target colored grid. The hidden rules make use of themes like objectness, goal-directedness, numbers & counting, basic geometry, and topology. Several input-output grid pairs are given as examples to help the system figure out the hidden rule in the puzzle, and no other information is given. Figure 1 shows three examples of ARC-AGI-1 training puzzles.

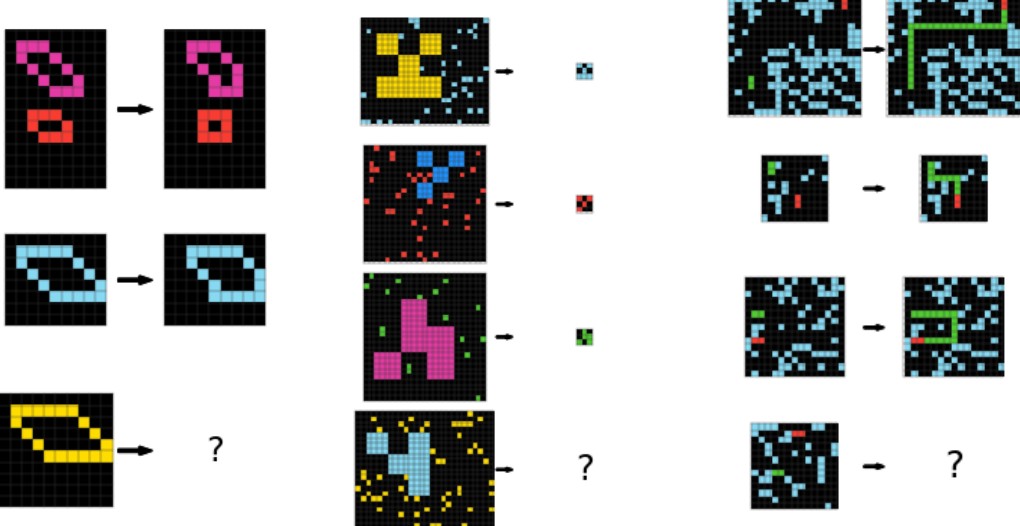

(a) **Hidden rule:** Shift every object right by one pixel, except the bottom/right edges of the object.

(b) **Hidden rule:** Shrink the big object and set its color to the scattered dots' color.

(c) **Hidden rule:** Extend the green line to meet the red line by turning when hitting a wall.

Figure 1: Three example ARC-AGI-1 puzzles.

While solutions based on LLMs built using internet scale data have scored 87.5% on this benchmark (Chollet, 2024), and neural networks trained on only ARC-AGI data have scored 40.3% (Wang et al., 2025), CompressARC takes the data-minimal perspective to its limits, opting to only train on the test puzzle.

Please refer to Appendix L for more details about the ARC-AGI-1 benchmark. An extended survey of other related work is also included in Appendix A. Note that we will generally refer to ARC-AGI-1 just as ARC-AGI in this paper.

## 3 METHOD

Occam's razor instructs that the shortest possible program whose return value matches the puzzle will likely also return the correct solution with it. Thus, we frame ARC-AGI as a code-golfing problem: to find the shortest possible program that reproduces the ARC-AGI dataset (Rissanen, 1978). In this case, the code must be entirely self-contained, receive no inputs, and must print out the entire ARC-AGI dataset of puzzles with any solutions filled in. Each puzzle takes the form of a tensor of shape $[\text{n\_example}, \text{width}, \text{height}, 2]$, containing color designations for every pixel in the $2 \times \text{n\_example}$ grids. Here, n_example counts the total number of input/output grid pairs in the puzzle, *including the test grid pair whose output grid is not known*. The shapes listed in this section are for explanatory purposes and the actual data format is introduced in Section 4. A naive first try at code-golfing the dataset may involve writing a program that hard-codes each puzzle in a giant string and prints it out. Improvements can be made by clever ways of de-duplicating structures in the printed data (e.g., introducing **for** loops, etc.); we will detail our own particular strategy below.

### 3.1 RESTRICTING THE PROGRAM SPACE

It is typically infeasible to run an algorithmic search to solve code-golf problems, because we would have to search through a huge number of increasingly lengthy programs to find one whose printout matches our requirements. Despite this, search can be made more amenable for our ARC-AGI code-golf problem if we restrict ourselves to a suitably well-conditioned subspace of programs. A formal derivation of CompressARC begins by picking a program subspace consisting of a template program (Algorithm 1) to be completed by substituting various hard-coded values into designated locations (shown in red). The template program generates each puzzle independently, and performs the operations for every puzzle:

1. it randomly samples a tensor $z$ of shape $[\text{n\_example}, \text{n\_colors}, \text{width}, \text{height}, 2]$ from a standard normal distribution, (line 4)

2. processes this with a neural network which outputs a $[\text{n\_example}, \text{n\_colors}, \text{width}, \text{height}, 2]$-shaped logit tensor, (line 6)

3. and obtains a $[\text{n\_example}, \text{width}, \text{height}, 2]$-shaped puzzle by sampling colors from the probability distribution implied by the logit tensor. (line 8)

---

**Algorithm 1:** Template for a short program that produce completed puzzles $P_{\text{filled}}$ with solutions filled in. Red text is to be substituted in with hard-coded values produced via Algorithm 2.

1  Define an equivariant_NN architecture;
2
3  **Set** seed_z = <seed_z$_1$>;                                    Hardcoded seed from Algo 2, puzzle 1
4  $z \leftarrow \text{sample}_{\text{seed\_z}}(N(0,1))$;                                    Generate inputs z
5  **Set** $\theta = <\theta_1>$;                                    Hardcoded weights from Algo 2, puzzle 1
6  grid_logits $\leftarrow$ equivariant_NN$_\theta(z)$;                                    Forward pass
7  **Set** seed_error = <seed_error$_1$>;                                    Hardcoded seed from Algo 2, puzzle 1
8  $P_{\text{filled}} \leftarrow \text{sample}_{\text{seed\_error}}(\text{grid\_logits})$;                                    Generate puzzle
9  **Print** $P_{\text{filled}}$
10
11  **Set** seed_z = <seed_z$_2$>;                                    Hardcoded seed from Algo 2, puzzle 2
12  (...code repeats for all puzzles)                                    ...

---

The template allows for two pseudo-random sampling seeds to be filled in for every puzzle (lines 3 and 7). The resulting printed puzzles can be guaranteed to match the true puzzles in the dataset by manipulating the final seed in line 7, which works after choosing any seed on line 3. With this guarantee in place, we can sum up the length of the code for the template, and find that the total length varies based on the number of bits/digits required to write down the two seeds. So, in order to search for short programs, we just need to make all the seeds as short as possible.

Multiple areas of the program can be adjusted to help minimize the seed length, and we will cover each in respective sections: the seeds and the weights on lines 3, 5, and 7 (Section 3.2 below), and the architecture on line 1 (Section 4).

## 3.2 SEED OPTIMIZATION

Algorithm 2 presents a method of optimizing the seeds and weights in template Algorithm 1 to reduce the total seed length. It first tries to manipulate the seed on line 3 of the template to imitate $z$ being sampled from a different learned normal distribution (line 8), and then tries to manipulate the second seed to guarantee matching puzzle output (line 11). It then performs gradient descent on the normal distribution parameters and the neural network weights to minimize the total seed length (lines 13-14).

---

**Algorithm 2:** Minimize Description Length, a.k.a. code-golf. n_example denotes the total number of input/output pairs in the puzzle, *including the test pair where the output is unknown.* Line 11: The smallest possible seed_error is picked so that the sampled puzzle $P_{\text{filled}}$ on line 12 matches the true puzzle $P$ on both the input and output grids of demonstration pairs, as well as the test inputs, but with no restriction on the test outputs.

1  **Input:** ARC-AGI dataset;
2  Define an equivariant_NN architecture;
3  **foreach** *puzzle P in ARC-AGI dataset* **do**
4     Randomly initialize weights $\theta$ for equivariant_NN$_\theta$;
5     Observe the dimensions n_example, n_colors, width, height of puzzle $P$;
6     Initialize input distribution: $\mu$ of shape [n_example, n_colors, width, height, 2], diagonal $\Sigma$;
7     **foreach** *step* **do**
8         **Set** seed_z, by manipulating to imitate $z \sim N(\mu, \Sigma)$;
9         $z \leftarrow \text{sample}_{\text{seed\_z}}(N(0, I), \text{shape}=[\text{n\_example, n\_colors, width, height, 2}])$;
10        grid_logits $\leftarrow$ equivariant_NN$_\theta(z)$;
11        **Set** seed_error, by manipulating to obtain desired puzzle $P$;
12        $P_{\text{filled}} \leftarrow \text{sample}_{\text{seed\_error}}(\text{grid\_logits})$;
13        $L \leftarrow \text{len(seed\_z)} + \text{len(seed\_error)}$;            $\approx$len(Algo 1) + C
14        $\mu, \Sigma, \theta \leftarrow \text{Adam}(\nabla_\mu L, \nabla_\Sigma L, \nabla_\theta L)$;
15     **end foreach**
16     Insert values seed_z, $\theta$, and seed_error into the pseudo-code for Algo 1;
17  **end foreach**
18  **Return** code for Algo 1

---

The idea of "manipulating the sampling seed for one distribution to imitate sampling from another distribution" tends to be fraught with technicalities and subtleties. Under favorable conditions, this is cleanly achievable through rejection sampling (Forsythe, 1972), which produces a sample from the imitated distribution using a seed whose expected length is the max log probability ratio between the two distributions. This can subsequently be improved and extended towards more general conditions using Relative Entropy Coding (REC) (Harsha et al., 2010; Havasi et al., 2018; Flamich et al., 2021), where the expected seed length lowers to the KL divergence between the two distributions, at the cost of only approximately achieving the desired sampling distribution. We refer readers to these references for details.

Our main issue with seed manipulation using REC within Algorithm 2 on lines 8 and 11 is that running

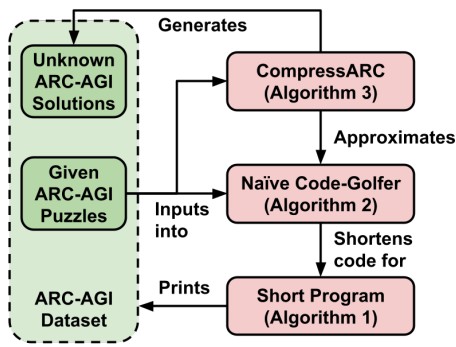

Figure 2: CompressARC approximates a specific code-golfing algorithm that converts the ARC-AGI puzzle dataset into the shortest program that prints it out exactly, along with any solutions. These printed solutions are assumed to be good predictors of the actual solutions, according to Occam's razor.

REC takes a lot of time when the imitated distribution is far from the sampling distribution (Flamich et al., 2021). So to make it faster, we skip these steps and imitate their expected downstream consequences instead, resulting in CompressARC (Algorithm 3). Namely for line 8, $z$ is now directly sampled from the imitated distribution, and the seed length from line 13 is replaced by its expected length, which is very close to the KL divergence between the imitated and sampling distributions according to the properties of REC (Flamich et al., 2021). For line 11, the unknown grids of the puzzle are sampled directly, and the seed length from line 13 is approximated closely by the negative log likelihood of sampling the known grids exactly, i.e., the crossentropy (see Appendix B for derivation).

---

**Algorithm 3:** CompressARC. It is the same as Algorithm 2, but with simulated seed manipulation, and truncated to return solved puzzles instead of description.

---

1  **Input:** ARC-AGI dataset;
2  Define an equivariant_NN architecture;
3  **foreach** *puzzle $P$ in ARC-AGI dataset* **do**
4     Randomly initialize weights $\theta$ for equivariant_NN$_\theta$;
5     Observe the dimensions n_example, n_colors, width, height of puzzle $P$;
6     Initialize input distribution: $\mu$ of shape [n_example, n_colors, width, height, 2], diagonal $\Sigma$;
7     **foreach** *step* **do**
8        $z \leftarrow \text{sample}(N(\mu, \Sigma))$;
9        grid_logits $\leftarrow$ equivariant_NN$_\theta(z)$;
10       $L \leftarrow \text{KL}(N(\mu, \Sigma) || N(0, I)) + \text{crossentropy}(\text{grid\_logits}, P)$;    $\approx$len(Algo 1) + C
11       $\mu, \Sigma, \theta \leftarrow \text{Adam}(\nabla_\mu L, \nabla_\Sigma L, \nabla_\theta L)$;
12    **end foreach**
13    $P_{\text{filled}} \leftarrow \text{sample}(\text{grid\_logits})$;
14    Add $P_{\text{filled}}$ to solved puzzles
15 **end foreach**
16 **Return** solved puzzles

---

Algorithm 3 (CompressARC) is now able to automatically code-golf the ARC-AGI dataset through successive refinement of template Algorithm 1, outputting solutions afterward. The only remaining component to specify is the neural network architecture used within template Algorithm 1, which we will design by hand. Since the architecture definition only appears once in Algorithm 1 while seeds appear repeatedly for every puzzle, using a sophisticated architecture can help us shorten the length of the template Algorithm 1, by trading off architecture description length to allow for shorter seeds. This serves as the primary motivation for us to heavily engineer the neural network architecture.

## 4 ARCHITECTURE

The central idea in designing the neural network architecture is to create a high probability of sampling the ARC-AGI puzzles, consequently reducing the length of the seeds and by extension the length of template Algorithm 1. According to the template structure, this means we need the neural network to have good inductive biases for transforming noise into reasonable-seeming ARC-AGI puzzles.[1]

Since ARC-AGI puzzles would be just as likely to appear in any combination of input/output example orderings, colors, orientations, etc., we want our network to assign them all equal probability by default. So, we made our architecture equivariant to example permutations, color permutations, rotations, and flips; (Cohen & Welling, 2016a) guaranteeing that computations applied to transformed inputs result in equivalently transformed outputs. For any asymmetries a puzzle may have, we require Algorithm 3 to manipulate the seed of the random input $z$, to bias the outputted puzzle one way or another.

The architecture, shown in Figure 3, consists of a decoding layer functioning like an embedding matrix (details in Appendix D.1), followed by a core with a residual backbone, followed by a linear readout on the channel dimension (see Appendix D.8). In the core, linear projections on the channel dimension read data from the residual into specially designed operations, which write their outputs

---

[1]The training puzzles play no role in our method other than to boost our efforts to better engineer the inductive biases into our layers.

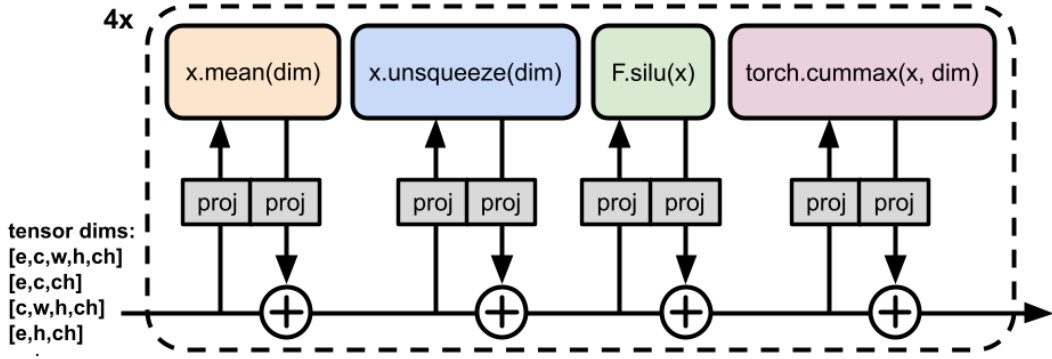

Figure 3: Core structure of CompressARC's neural network, which operates on multitensor data. Individual operations (colored) read and write to a residual backbone through learned projections (grey) in the channel dimension. The network is equivariant to permutations of indices along the other, non-channel dimensions as a result. Some layers like cummax break certain geometric symmetries, giving the architecture specific geometric abilities listed in Appendix H. Normalization, softmax, shift, and directional layers are not shown.

back into the residual through another projection. Normalization operations are scattered throughout the layers, and then the whole block of core layers is repeated 4 times. This is much like a transformer architecture, (Vaswani et al., 2023) except that the specially designed operations are not the attention and nonlinear operations on sequences, but instead the following operations on puzzle-shaped data:

- summing one tensor along an axis and/or broadcasting the result back up, (see Appendix D.2)
- taking the softmax along one or multiple axes of a tensor, (see Appendix D.3)
- taking a cumulative max along one of the geometric dimensions of a tensor, (see Appendix D.4)
- shifting by one pixel along one of the geometric dimensions of a tensor, (see Appendix D.4)
- elementwise nonlinearity, (see Appendix D.6)
- normalization along the channel dimension, (see Appendix D.7)

along with one more described in Appendix D.5. The operations have no parameters and have their behaviors controlled by their residual read/write weights. All of these read/write projections operate on the channel dimension. We used 16 channels in some parts of the backbone and 8 in others to reduce computational costs. Since these projections take up the majority of the model weights, the entire model only has 76K parameters.

**The actual data format** that the neural network uses for computation is not a single tensor shaped like [n_example, n_colors, width, height, channel], but instead a bucket of tensors that each have a different subset of these dimensions, for example a [n_colors, width, channel] tensor. Both the input $z$ to the network and the outputted logits, as well as all of the internal activations, take the form of a multitensor. Generally, there is a tensor for every subset of these dimensions for storing information of that shape, which helps to build useful inductive biases. For example, an assignment of grid columns to colors can be stored as a one-hot table in the [color, width, channel]-shaped tensor. More details on multitensors are in Appendix C.

## 5 RESULTS

We gave CompressARC 2000 inference-time training steps on every puzzle, taking about 20 minutes per puzzle. CompressARC correctly solved 20% of evaluation set puzzles and 34.75% of training set puzzles within this budget of inference-time compute. Figure 4 shows the performance increase as more inference-time compute is given. Tables 4 and 5 in the Appendix document the numerical solve accuracies with timings.

Table 1: Comparison of solution accuracies of various methods for ARC-AGI-1, sorted by the amount of training data used. Each method makes two solution guesses per puzzle, and a guess is only correct if the grid shape and pixel colors are all correct. The U-Net (Ronneberger et al., 2015) baseline is a supervised model trained during inference time on only the demonstration input/output pairs of grids in the test puzzle to match the constraints of CompressARC; details in Appendix I.

| Method | Trained on: | Neural | Acc. | Dataset split |
|--------|-------------|--------|------|---------------|
| Random guessing | Nothing | ✗ | 0% | All |
| Brute force rule search (Kamradt, 2024) | Nothing | ✗ | 40% | Private Eval |
| U-Net baseline | Target puzzle | ✔ | 0.75% | Public Eval |
| **CompressARC (ours)** | **Target puzzle** | ✔ | **20%** | **Public Eval** |
| HRM ablation (ARC Prize Team, 2025) | Test puzzles | ✔ | 31% | Public Eval |
| HRM (Wang et al., 2025) | Train+test puzzles | ✔ | 40.3% | Public Eval |
| OpenAI o3 high (Chollet, 2024) | Internet scale data | ✔ | 87.5% | Semi-Priv. Eval |

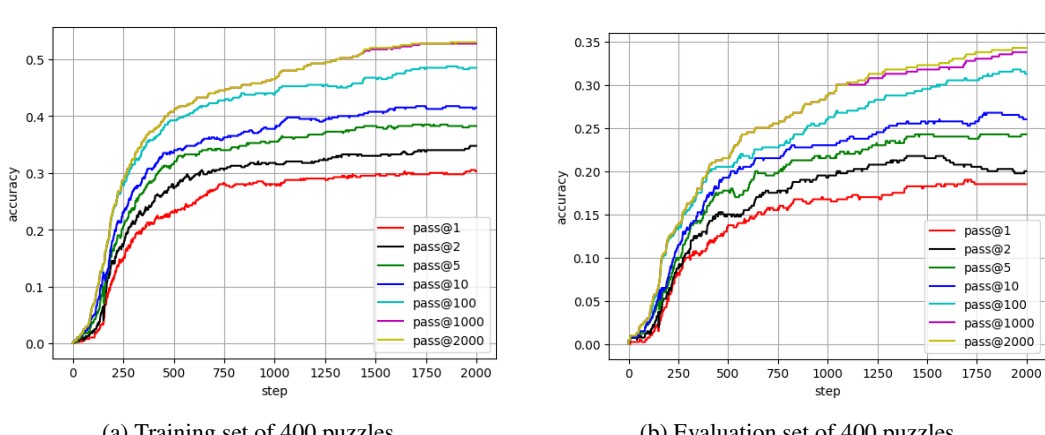

(a) Training set of 400 puzzles.

(b) Evaluation set of 400 puzzles.

Figure 4: CompressARC's puzzle solve accuracy rises as inference time learning progresses. Various numbers of allowed solution guesses (pass@n) for accuracy measurement are shown. The official benchmark is reported with 2 allowed guesses, which is why we report 20% on the evaluation set.

### 5.1 WHAT PUZZLES CAN AND CAN'T WE SOLVE?

**CompressARC tries to use its abilities to figure out as much as it can, until it gets bottlenecked by one of it's inabilities.**

For example, puzzle 28e73c20 in the training set requires extension of a pattern from the edge towards the middle, as shown in Figure 12a in the Appendix. Given the layers in it's network, CompressARC is generally able to extend patterns for short ranges but not long ranges. So, it does the best that it can, and correctly extends the pattern a short distance before guessing at what happens near the center (Figure 12b, Appendix). Appendix H includes a list of which abilities we have empirically seen CompressARC able to and not able to perform.

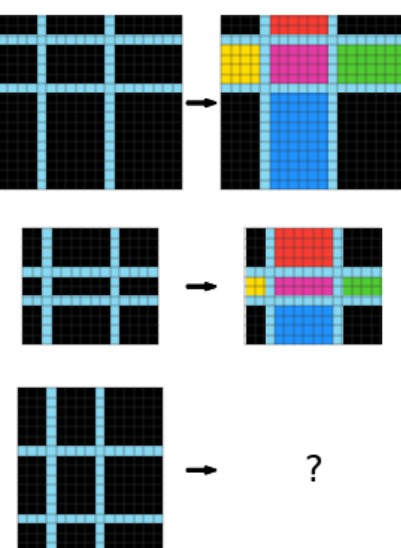

Figure 5: Color the Boxes, puzzle 272f95fa.

### 5.2 CASE STUDY: COLOR THE BOXES

In the puzzle shown (Figure 5), you must color the boxes depending on which side of the grid the box is on. We call this puzzle "Color the Boxes".

**Human Solution:** We first realize that the input is divided into boxes, and the boxes are still there in the output, but now they're colored. We then try to figure out which colors go in which boxes. First, we notice that the corners are always black. Then, we notice that the middle is always magenta. And after that, we notice that the color of the side boxes depends on which direction they are in: red for up, blue for down, green for right, and yellow for left. At this point, we copy the input over to the answer grid, then we color the middle box magenta, and then color the rest of the boxes according to their direction.

**CompressARC Solution:** Table 2 shows CompressARC's learning behavior over time. After CompressARC is done learning, we can deconstruct its learned $z$ distribution to find that it codes for a color-direction correspondence table and row/column divider positions (Figure 7).

During training, the reconstruction error fell extremely quickly. It remained low on average, but would spike up every once in a while, causing the KL from $z$ to bump upwards at these moments, as shown in Figure 6a.

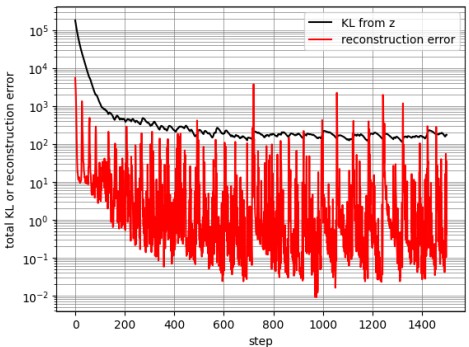
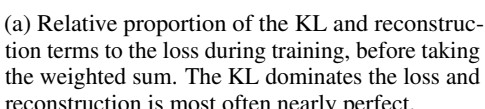
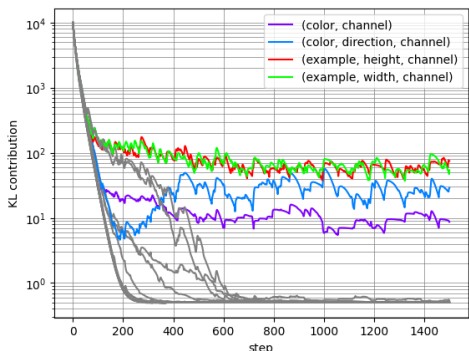

(a) Relative proportion of the KL and reconstruction terms to the loss during training, before taking the weighted sum. The KL dominates the loss and reconstruction is most often nearly perfect.

(b) Breaking down the KL loss during training into contributions from each individual shaped tensor in the multitensor $z$. Four tensors dominate, indicating they contain information, and the other 14 fall to zero, indicating their lack of information content.

Figure 6: Breaking down the loss components during training tells us where and how CompressARC prefers to store information relevant to solving a puzzle.

### 5.2.1 SOLUTION ANALYSIS

We observe the representations stored in $z$ to see how CompressARC learns to solve Color the Boxes.

Since $z$ is a multitensor, each of the tensors it contains produces an additive contribution to the total KL for $z$. By looking at the per-tensor contributions (see Figure 6b), we can determine which tensors in $z$ code for information that is used to represent the puzzle.

All the tensors fall to zero information content during training, except for four tensors. In some replications of this experiment, we saw one of these four necessary tensors fall to zero information content, and CompressARC typically does not recover the correct answer after that. Here we are showing a lucky run where the [color, direction, channel] tensor almost falls but gets picked up 200 steps in, which is right around when the samples from the model begin to show the correct colors in the correct boxes.

We can look at the average output of the decoding layer (explained in Appendix D.1) corresponding to individual tensors of $z$, to see what information is stored there (see Figure 7). Each tensor contains a vector of dimension n_channels for various indices of the tensor. Taking the PCA of these vectors reveals some number of activated components, telling us how many pieces of information are coded by the tensor.

Table 2: CompressARC learning the solution for Color the Boxes, over time.

| Learning steps | What is CompressARC doing? | Sampled solution guess |
|---|---|---|
| 50 | CompressARC's network outputs an answer grid (sample) with light blue rows/columns wherever the input has the same. It has noticed that all the other input-output pairs in the puzzle exhibit this correspondence. It doesn't know how the other output pixels are assigned colors; an exponential moving average of the network output (sample average) shows the network assigning mostly the same average color to non-light-blue pixels. |  |
| 150 | The network outputs a grid where nearby pixels have similar colors. It has likely noticed that this is common among all the outputs, and is guessing that it applies to the answer too. |  |
| 200 | The network output now shows larger blobs of colors that are cut off by the light blue borders. It has noticed the common usage of borders to demarcate blobs of colors in other outputs, and applies the same idea here. It has also noticed black corner blobs in other given outputs, which the network imitates. |  |
| 350 | The network output now shows the correct colors assigned to boxes of the correct direction from the center. It has realized that a single color-to-direction mapping is used to pick the blob colors in the other given outputs, so it imitates this mapping. It is still not the best at coloring within the lines, and it is also confused about the center blob, probably because the middle does not correspond to a direction. Nevertheless, the average network output does show a tinge of the correct magenta color in the middle, meaning the network is catching on. |  |
| 1500 | The network is as refined as it will ever be. Sometimes it will still make a mistake in the sample it outputs, but this uncommon and filtered out. |  |

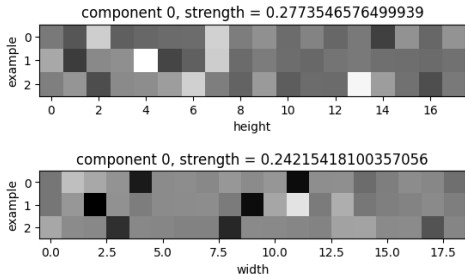

(a) **(example, height, channel) and (example, width, channel) tensors.** For every example and row/column, there is a vector of dimension n_channels. Taking the PCA of this set of vectors, the top principal component (>1000 times stronger than the other components for both tensors) visualized as the (example, height) and (example, width) matrices shown above tells us which example/row and example/column combinations are uniquely identified by the stored information. **For every example, the two brightest pixels in the top matrix give positions of the light blue rows in the puzzle grids, and the darkest two pixels in the bottom matrix indicate the columns.**

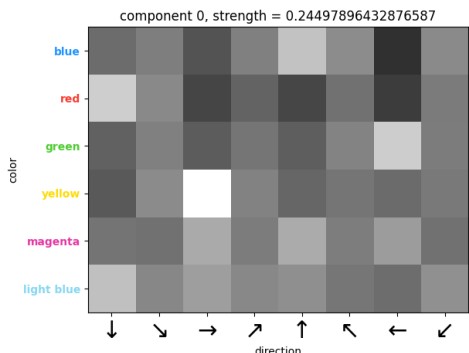

(b) (**direction**, **color**, **channel**) **tensor.** A similar style PCA decomposition: the graph shows the top principal component for this tensor. The four brightest pixels identify blue with up, green with left, red with down, and yellow with right. **This tensor tells each direction which color to use for the opposite edge's box.** The top principal component is 829 times stronger than the next principal component.

Figure 7: Breaking down the loss components during training tells us where and how CompressARC prefers to store information relevant to solving a puzzle.

## 6 DISCUSSION

The prevailing reliance of modern deep learning on high-quality data has put the field in a chokehold when applied to problems requiring intelligent behavior that have less data available. This is especially true for the data-limited ARC-AGI benchmark, where LLMs trained on specially augmented, extended, and curated datasets dominate (Knoop, 2024). In the midst of this circumstance, we built CompressARC, which not only uses no training data at all, but forgoes the entire process of pretraining altogether. One should intuitively expect this to fail and solve no puzzles at all, but by applying MDL to the target puzzle during inference time, CompressARC solves a surprisingly large portion of ARC-AGI-1.

CompressARC's theoretical underpinnings come from minimizing the length of a programmatic description of the target puzzle. While other MDL search strategies have been scarce due to the intractablly large search space of possible programs, CompressARC explores a simplified, neural network-based search space through gradient descent. Though CompressARC's architecture is heavily engineered, its incredible ability to generalize from as low as two demonstration input/output pairs puts it in an entirely new regime of generalization for ARC-AGI.

Efficiency improvement remains a valuable direction for future work on CompressARC. CompressARC makes use of many custom operations (See Appendices C and D), and adding JIT-compiled kernels or fused CUDA kernels would increase the training iteration speed. Improvements will naturally have a larger effect on larger grids since our architecture's runtime scales with the number of pixels in the puzzle.

We challenge the assumption that intelligence must arise from massive pretraining and data, showing instead that clever use of MDL and compression principles can lead to surprising capabilities. We use CompressARC as a proof of concept to demonstrate that modern deep learning frameworks can be melded with MDL to create a possible alternative, complimentary route to AGI.

Note: Large Language Models (LLMs) were used to polish the writing in this paper, in particular to find the most clear and concise way of introducing our work in Section 1.

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

## A RELATED WORK

### A.1 EQUIVALENCE OF COMPRESSION AND INTELLIGENCE

The original inspiration of this work came from the Hutter Prize (Hutter, 2006), which awards a prize for those who can compress a file of Wikipedia text the most, as a motivation for researchers to build intelligent systems. It is premised upon the idea that the ability to compress information is equivalent to intelligence.

This equivalence between intelligence and compression has a long history. For example, when talking about intelligent solutions to prediction problems, the ideal predictor implements Solomonoff Induction, a theoretically best possible but uncomputable prediction algorithm that works universally for all prediction tasks (Solomonoff, 1964). This prediction algorithm is then equivalent to a best possible compression algorithm whose compressed code length is the Kolmogorov Complexity of the data (Kolmogorov, 1998). This prediction algorithm can also be used to decode a description of the data of minimal length, linking these formulations of intelligence to MDL (Rissanen, 1978). In our work, we try to approximate this best possible compression algorithm with a neural network.

### A.2 INFORMATION THEORY AND CODING THEORY

Since we build an information compression system, we make use of many results in information theory and coding theory. The main result required to motivate our model architecture is the existence of Relative Entropy Coding (REC) (Flamich et al., 2021). The fact that REC exists means that as long as a KL divergence can be bounded, the construction of a compression algorithm is always possible and the issue of realizing the algorithm can be abstracted away. Thus, problems about coding theory and translating information from Gaussians into binary and back can be ignored, since we can figure out the binary code length directly from the Gaussians instead. In other words, we only need to do enough information theory using the Gaussians to get the job done, with no coding theory at all. While the existence of arithmetic coding (Langdon, 1984) would suffice to abstract the problem away when distributions are discrete, neural networks operate in a continuous space so we need REC instead.

Our architecture sends $z$ information through an additive white Gaussian noise (AWGN) channel, so the AWGN channel capacity formula (Gaussian input Gaussian noise) plays a heavy role in the design of our decoding layer (Shannon, 1948).

### A.3 VARIATIONAL AUTOENCODERS

The decoder side of the variational autoencoder (Kingma & Welling, 2022) serves as our decompression algorithm. While we would use something that has more general capabilities like a neural Turing machine (Graves et al., 2014) instead, neural Turing machines are not very amenable to gradient descent-based optimization so we stuck with the VAE.

VAEs have a long history of developments that are relevant to our work. At one point, we tried using multiple decoding layers to make a hierarchical VAE decoder (Sønderby et al., 2016) instead. This does not affect the KL calculation because a channel capacity with feedback is equal to the channel capacity without feedback (Shannon, 1956). But, we found empirically that the first decoding layer would absorb all of the KL contribution, making the later decoding layers useless. Thus, we only used one decoding layer at the beginning.

The beta-VAE (Higgins et al., 2017) introduces a reweighting of the reconstruction loss to be stronger than the KL loss, and we found that to work well in our case. The NVAE applies a non-constant weighting to loss components (Vahdat & Kautz, 2021). A rudimentary form of scheduled loss recombination is used in CompressARC.

### A.4 OTHER ARC-AGI METHODS

Top-scoring methods to solve ARC-AGI rely on converting puzzle grids into text and then feeding them into a pretrained large language model which is prompted to find the solution. The predominant techniques involve either using the LLM to output the solution grid directly (Li et al., 2024b; Cole

& Osman, 2025; Akyürek et al., 2024), or output a program that can be run to manipulate the grids instead(Li et al., 2024b; Greenblatt, 2024; Barbadillo, 2025; Berman, 2024). Oftentimes, these methods employ several tricks to improve performance:

- Fine-tuning on training puzzle data
  - Applying data augmentation to increase the effective number of puzzles to fine-tune on (Akyürek et al., 2024)
  - Fine-tuning on synthetic data (Li et al., 2024a; Akyürek et al., 2024)
- Employing inference-time training approaches
  - Fine-tuning an individual model specific to each test puzzle, during test time (Akyürek et al., 2024)
  - Test-time training (TTT) techniques (Sun et al., 2020; Barbadillo, 2024)
- Sampling many model outputs or random augmentations of the test puzzle, for ensembling (Cole & Osman, 2025; Greenblatt, 2024)
- LLM reasoning (Chollet, 2024)

Such methods have managed to score up to 87.5% on the semi-private split of ARC-AGI, at a cost of over $200 equivalent of inference-time compute per puzzle (Chollet, 2024). These approaches all make use of language models that were pretrained on the entire internet, which is in contrast to CompressARC, whose only training data is the test puzzle. Their commonalities with CompressARC are mainly in the emphasis on training individual models on individual test puzzles and the use of ensembling to improve solution predictions.

Aside from these methods, several other methods have been studied:

- An older class of methods consists of hard-coded, large-scale searches through program spaces in hand-written domain-specific languages designed specifically for ARC (Hodel, 2024; Odouard, 2024). While these methods do not use neural networks to solve puzzles and are less instructive towards the field of machine learning, they share the commonality of using heavily engineered components designed specifically for ARC-AGI.
- (Bonnet & Macfarlane, 2024) introduced a VAE-based method for searching through a latent space of programs. This is the most similar work to ours that we found due to their VAE setup.

### A.5  Deep Learning Architectures

We designed our own neural network architecture from scratch, but not without borrowing crucial design principles from many others.

Our architecture is fundamentally structured like a transformer, consisting of a residual stream where representations are stored and operated upon, followed by a linear head (Vaswani et al., 2023; He et al., 2015). Pre-and post-norms with linear up- and down-projections allow layers to read and write to the residual stream (Xiong et al., 2020). The SiLU-based nonlinear layer is especially similar to a transformer's (Hendrycks & Gimpel, 2023).

Our equivariance structures are inspired by permutation-invariant neural networks, which are a type of equivariant neural network (Zaheer et al., 2018; Cohen & Welling, 2016b). Equivariance transformations are taken from common augmentations to ARC-AGI puzzles.

## B  Seed Length Estimation by KL and CrossEntropy

In Section 3.1, we estimate $\text{len}(\text{seed\_z})$ in line 12 of template Algorithm 2 as $\text{KL}(N(\mu, \Sigma)||N(0, I))$, and $\text{len}(\text{seed\_error})$ as $\text{crossentropy}(\text{grid\_logits}, P)$. In this section, we will argue for the reasonability of this approximation. Readers may also refer to Flamich et al. (2021), which introduces a better seed manipulation method as "Relative Entropy Coding" (REC). Flamich et al. (2021) shows that seed communication is effectively the most bit-efficient way for an encoder and decoder to communicate samples from a distribution $Q$ if there is a shared source of randomness $P$. This uses nearly $\text{KL}(P||Q)$ expected bits per sample communicated. We urge readers to refer to Flamich

et al. (2021) for details regarding the manipulation procedure, runtime and memory analysis, and approximation strength. Below, we follow with our own effort at reasoning through why.

To recap, we original procedure in Algorithm 2 manipulates the seed for sampling $z \sim N(0, I)$ to simulate as though $z \sim N(\mu, \Sigma)$, and we would like to show that we can closely approximate this sampling using an expected number of seed bits close to $\text{KL}(N(\mu, \Sigma)||N(0, I))$.

For sake of illustration, suppose for instance that Algorithm 2 implements something similar to rejection sampling, (Forsythe, 1972) iterating through seeds one by one and accepting the sample with probability $\min(1, cw(z))$ for some $c \ll 1$, where $w(z)$ is the probability ratio

$$w(z) = \frac{N(z; \mu, \Sigma)}{N(z; 0, I)}$$

When we pick a small enough $c$, the sampling distribution becomes arbitrarily close to $N(\mu, \Sigma)$ as desired. With this $c$, we would like to show that the expected number of rejections leads us to end up with a seed length close to the KL.

We would first like to lower bound the probability $P_{\text{accept}}$ of accepting at each step, which is

$$P_{\text{accept}} = \int N(z; 0, I)\min(1, cw(z))\, \mathrm{d}z$$

$$= \int N(z; 0, I)\min\left(1, \frac{cN(z; \mu, \Sigma)}{N(z; 0, I)}\right)\, \mathrm{d}z$$

$$= \int \min\left(N(z; 0, I), cN(z; \mu, \Sigma)\right)\, \mathrm{d}z$$

We will follow a modified version of a derivation of the Bretagnolle–Huber inequality (Bretagnolle & Huber, 1978) by Tsybakov (2008) to derive a bound on the KL:

$$(1 + c)P_{\text{accept}} \geq (1 + c - P_{\text{accept}})P_{\text{accept}}$$

$$= \left(\int \max\left(N(z; 0, I), cN(z; \mu, \Sigma)\right)\, \mathrm{d}z\right)\left(\int \min\left(N(z; 0, I), cN(z; \mu, \Sigma)\right)\, \mathrm{d}z\right)$$

where applying the Cauchy-Schwarz inequality with a function space inner product,

$$\geq \left(\int \sqrt{\max\left(N(z; 0, I), cN(z; \mu, \Sigma)\right)\min\left(N(z; 0, I), cN(z; \mu, \Sigma)\right)}\, \mathrm{d}z\right)^2$$

$$= \left(\int \sqrt{cN(z; 0, I)N(z; \mu, \Sigma)}\, \mathrm{d}z\right)^2$$

$$= c\exp\left(2\ln\int \sqrt{N(z; 0, I)N(z; \mu, \Sigma)}\, \mathrm{d}z\right)$$

$$= c\exp\left(2\ln\int N(z; \mu, \Sigma)\sqrt{\frac{N(z; 0, I)}{N(z; \mu, \Sigma)}}\, \mathrm{d}z\right)$$

$$= c\exp\left(2\ln\mathbb{E}_{z \sim N(z; \mu, \Sigma)}\left[\sqrt{\frac{N(z; 0, I)}{N(z; \mu, \Sigma)}}\right]\right)$$

and following with Jensen's inequality,

$$\geq c\exp\left(\mathbb{E}_{z \sim N(z; \mu, \Sigma)}\left[\ln\frac{N(z; 0, I)}{N(z; \mu, \Sigma)}\right]\right)$$

$$= c\exp\left(-\text{KL}(N(\mu, \Sigma)||N(0, I))\right)$$

leads to an acceptance probability of at least

$$P_{\text{accept}} \geq \frac{c}{1 + c}\exp\left(-\text{KL}(N(\mu, \Sigma)||N(0, I))\right)$$

Therefore, according to the rejection sampling procedure, the number of samples proposed (i.e. the expected seed) is at most the inverse of this acceptance probability,

$$\text{seed\_z} \leq \frac{1+c}{c} \exp(\text{KL}(N(\mu, \Sigma)||N(0,1)))$$

so the expected seed length is at most around the logarithm,

$$\text{len(seed\_z)} \leq \text{KL}(N(\mu, \Sigma)||N(0,1)) + \log(1+c) - \log c$$

matching up with our stated KL approximation of the seed length.

For the seed_error term, Algorithm 2 manipulates the seed to sample a puzzle $P$ from a distribution implied by some logits. This is effectively the same as sampling grid_logits $\sim$ Categorical_distribution(logits) and manipulating the seed to try to get grid_logits $\sim$ Delta_distribution($P$). Then, the same KL-based bound on required seed length can be used once again. The expected seed_error length is at most

$$\text{KL}(\text{Delta\_distribution}(P)||\text{Categorical\_distribution(logits)}) + \log(1+c) - \log c$$

which simplifies as

$$\mathbb{E}_{x \sim \text{Delta\_distribution}(P)} \left[ \log \frac{\delta(x = P)}{\text{Categorical\_probability}(x; \text{logits})} \right] + \log(1+c) - \log c$$

$$= \log \frac{\delta(P = P)}{\text{Categorical\_probability}(P; \text{logits})} + \log(1+c) - \log c$$

$$= -\log \text{Categorical\_probability}(P; \text{logits}) + \log(1+c) - \log c$$

$$= \text{cross\_entropy}(\text{logits}, P) + \log(1+c) - \log c$$

where the $\delta$ is 1 when the statement within is true, and 0 otherwise.

## C MULTITENSORS

The actual data ($z$, hidden activations, and puzzles) passing through our layers comes in a format that we call a "**multitensor**", which is just a bucket of tensors of various shapes, as shown in Figure 8. All the equivariances we use can be described in terms of how they change a multitensor.

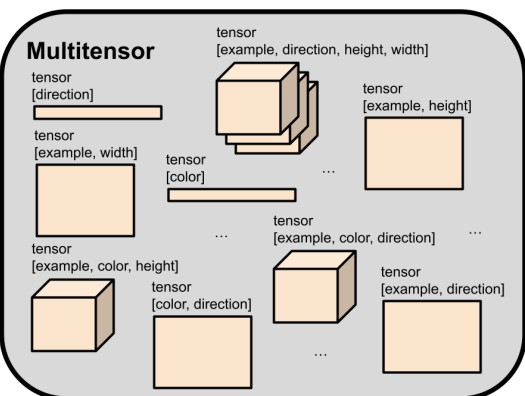

Figure 8: Our neural network's internal representations come in the form of a "multitensor", a bucket of tensors of different shapes. One of the tensors is shaped like [example, color, height, width, channel], an adequate shape for storing a whole ARC-AGI puzzle.

Most common classes of machine learning architectures operate on a single type of tensor with constant rank. LLMs operate on rank-3 tensors of shape [n_batch, n_tokens, n_channels], and Convolutional Neural Networks (CNNs) operate on rank-4 tensors of shape [n_batch, n_channels, height, width]. Our multitensors are a set of varying-rank tensors of unique type, whose dimensions are a subset of a rank-6 tensor of shape

[n_example, n_colors, n_directions, height, width, n_channels], as illustrated in Figure 8. We always keep the channel dimension, so there are at most 32 tensors in each multitensor. We also maintain several rules (see Appendix E.1) that determine whether a tensor shape is "legal" or not, which reduces the number of tensors in a multitensor to 18.

| Dimension | Description |
|-----------|-------------|
| Example | Number of examples in the ARC-AGI puzzle, including the one with held-out answer |
| Color | Number of unique colors in the ARC-AGI puzzle, not including black, see Appendix F.2 |
| Direction | 8 |
| Height | Determined when preprocessing the puzzle, see Appendix F.1 |
| Width | Determined when preprocessing the puzzle, see Appendix F.1 |
| Channel | In the residual connections, the size is 8 if the direction dimension is included, else 16. Within layers it is layer-dependent. |

Table 3: Size conventions for multitensor dimensions.

To give an idea of how a multitensor stores data, an ARC-AGI puzzle can be represented by using the [example, color, height, width, channel] tensor, by using the channel dimension to select either the input or output grid, and the height/width dimensions for pixel location, a one hot vector in the color dimension, specifying what color that pixel is. The [example, height, channel] and [example, width, channel] tensors can similarly be used to store masks representing grid shapes for every example for every input/output grid. All those tensors are included in a single multitensor that is computed by the network just before the final linear head (described in Appendix D.8).

When we apply an operation on a multitensor, we by default assume that all non-channel dimensions are treated identically as batch dimensions by default. The operation is copied across the indices of dimensions unless specified. This ensures that we keep all our symmetries intact until we use a specific layer meant to break a specific symmetry.

A final note on the channel dimension: usually when talking about a tensor's shape, we will not even mention the channel dimension as it is included by default.

# D LAYERS IN THE ARCHITECTURE

## D.1 DECODING LAYER

This layer's job is to sample a multitensor $z$ and bound its information content, before it is passed to the next layer. This layer and outputs the KL divergence between the learned $z$ distribution and $N(0, I)$. Penalizing the KL prevents CompressARC from learning a distribution for $z$ that memorizes the ARC-AGI puzzle in an uncompressed fashion, and forces CompressARC to represent the puzzle more succinctly. Specifically, it forces the network to spend more bits on the KL whenever it uses $z$ to break a symmetry, and the larger the symmetry group broken, the more bits it spends.

This layer takes as input:

- A learned target multiscalar, called the "target capacity".[2] The decoding layer will output $z$ whose information content per tensor is close to the target capacity,[3]

- learned per-element means for $z$,[4]

- learned per-element capacity adjustments for $z$.

---

[2]Target capacities are exponentially parameterized and rescaled by 10x to increase sensitivity to learning, initialized at a constant $10^4$ nats per tensor, and forced to be above a minimum value of half a nat.

[3]The actual information content, which the layer computes later on, will be slightly different because of the per-element capacity adjustments.

[4]Means are initialized using normal distribution of variance $10^{-4}$.

We begin by normalizing the learned per-element means for $z$.[5] Then, we figure out how much Gaussian noise we must add into every tensor to make the AWGN channel capacity (Shannon, 1948) equal to the target capacity for every tensor (including per-element capacity adjustments). We apply the noise to sample $z$, keeping unit variance of $z$ by rescaling.[6]

We compute the information content of $z$ as the KL divergence between the distribution of this sample and $N(0, 1)$.

Finally, we postprocess the noisy $z$ by scaling it by the sigmoid of the signal-to-noise ratio.[7] This ensures that $z$ is kept as-is when its variance consists mostly of useful information and it is nearly zero when its variance consists mostly of noise. All this is done 4 times to make a channel dimension of 4. Then we apply a projection (with different weights per tensor in the multitensor, i.e., per-tensor projections) mapping the channel dimension up to the dimension of the residual stream.

### D.2 MULTITENSOR COMMUNICATION LAYER

This layer allows different tensors in a multitensor to interact with each other.

First, the input from the residual stream passes through per-tensor projections to a fixed size (8 for downwards communication and 16 for upwards communication). Then a message is sent to every other tensor that has at least the same dimensions for upwards communication, or at most the same dimensions for downwards communication. This message is created by either taking means along dimensions to remove them, or unsqueezing+broadcasting dimensions to add them, as in Figure 9. All the messages received by every tensor are summed together and normalization is applied. This result gets up-projected back and then added to the residual stream.

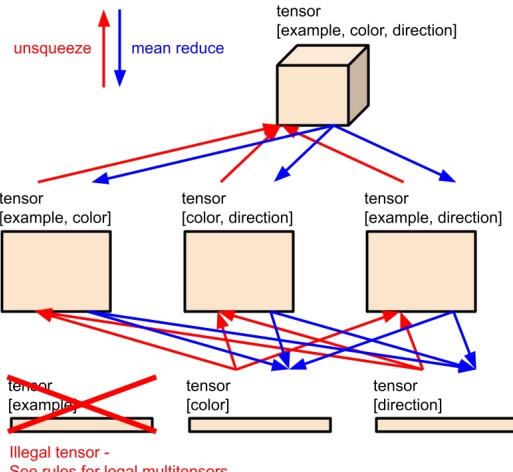

Figure 9: Multitensor communication layer. Higher rank tensors shown at the top, lower rank at the bottom. Tensors transform between ranks by mean reduction and unsqueezing dimensions.

### D.3 SOFTMAX LAYER

This layer allows the network to work with internal one-hot representations, by giving it the tools to denoise and sharpen noisy one-hot vectors. For every tensor in the input multitensor, this layer lists out all the possible subsets of dimensions of the tensor to take a softmax over,[8] takes the softmax

---

[5]Means and variances for normalization are computed along all non-channel dimensions.

[6]There are many caveats with the way this is implemented and how it works; please refer to the code (see Appendix O) for more details.

[7]We are careful not to let the postprocessing operation, which contains unbounded amounts of information via the signal-to-noise ratios, to leak lots of information across the layer. We only let a bit of it leak by averaging the signal-to-noise ratios across individual tensors in the multitensor.

[8]One exception: we always include the example dimension in the subset of dimensions.

over these subsets of dimensions, and concatenates all the softmaxxed results together in the channel dimension. The output dimension varies across different tensors in the multitensor, depending on their tensor rank. A pre-norm is applied, and per-tensor projections map to and from the residual stream. The layer has input channel dimension of 2.

### D.4 Directional Cummax/Shift Layer

The directional cummax and shift layers allow the network to perform the non-equivariant cummax and shift operations in an equivariant way, namely by applying the operations once per direction, and only letting the output be influenced by the results once the directions are aggregated back together (by the multitensor communication layer). These layers are the sole reason we included the direction dimension when defining a multitensor: to store the results of directional layers and operate on each individually. Of course, this means when we apply a spatial equivariance transformation, we must also permute the indices of the direction dimension accordingly, which can get complicated sometimes.

The directional cummax layer takes the eight indices of the direction dimension, treats each slice as corresponding to one direction (4 cardinal, 4 diagonal), performs a cumulative max in the respective direction for each slice, does it in the opposite direction for half the channels, and stacks the slices back together in the direction dimension. An illustration is in Figure 10. The slices are rescaled to have min $-1$ and max $1$ before applying the cumulative max.

The directional shift layer does the same thing, but for shifting the grid by one pixel instead of applying the cumulative max, and without the rescaling.

Some details:

- Per-tensor projections map to and from the residual stream, with pre-norm.
- Input channel dimension is 4.
- These layers are only applied to the $[example, color, direction, height, width, channel]$ and $[example, direction, height, width, channel]$ tensors in the input multitensor.

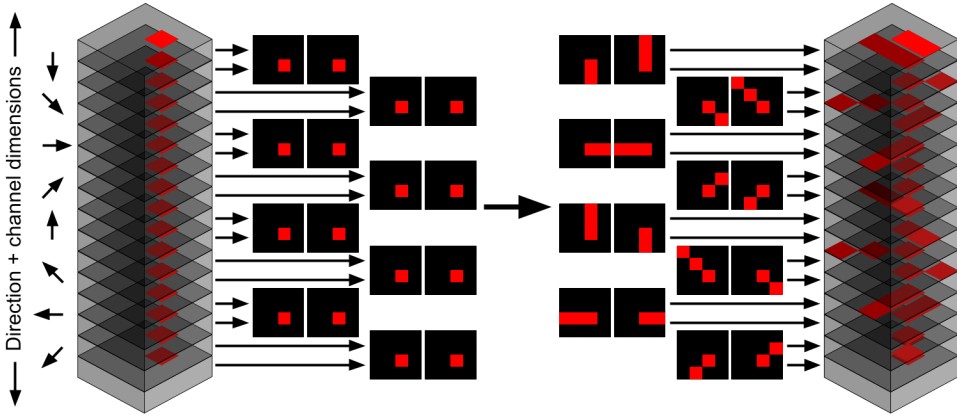

Figure 10: The directional cummax layer takes a directional tensor, splits it along the direction axis, and applies a cumulative max in a different direction for each direction slice. This operation helps CompressARC transport information across long distances in the puzzle grid.

### D.5 Directional Communication Layer

By default, the network is equivariant to permutations of the eight directions, but we only want symmetry up to rotations and flips. So, this layer provides a way to send information between two slices in the direction dimension, depending on the angular difference in the two directions. This layer defines a separate linear map to be used for each of the 64 possible combinations of angles,

but the weights of the linear maps are minimally tied such that the directional communication layer is equivariant to reflections and rotations. This gets complicated really fast, since the direction dimension's indices also permute when equivariance transformations are applied. Every direction slice in a tensor accumulates its 8 messages, and adds the results together.[9]

For this layer, there are per-tensor projections to and from the residual stream with pre-norm. The input channel dimension is 2.

### D.6 NONLINEAR LAYER

We use a SiLU nonlinearity with channel dimension 16, surrounded by per-tensor projections with pre-norm.

### D.7 NORMALIZATION LAYER

We normalize all the tensors in the multitensor, using means and variances computed across all dimensions except the channel dimension. Normalization as used within other layers also generally operates this way.

### D.8 LINEAR HEADS

We must take the final multitensor, and convert it to the format of an ARC-AGI puzzle. More specifically, we must convert the multitensor into a distribution over ARC-AGI puzzles, so that we can compute the log-likelihood of the observed grids in the puzzle.

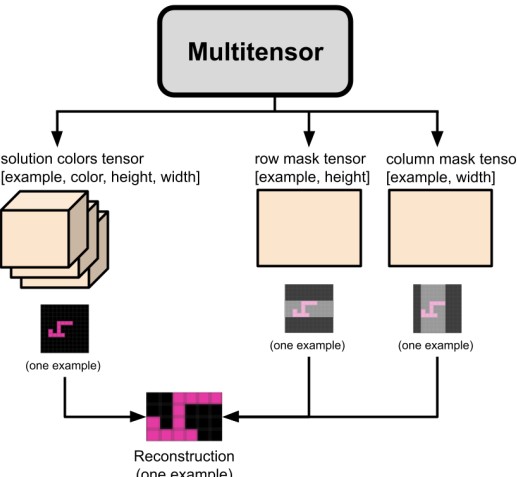

Figure 11: The linear head layer takes the final multitensor of the residual stream and reads a $[\text{example}, \text{color}, \text{height}, \text{width}, \text{channel}]$ tensor to be interpreted as color logits, and a $[\text{example}, \text{height}, \text{channel}]$ tensor and a $[\text{example}, \text{width}, \text{channel}]$ tensor to serve as shape masks.

The colors of every pixel for every example for both input and output, have logits defined by the $[\text{example}, \text{color}, \text{height}, \text{width}, \text{channel}]$ tensor, with the channel dimension linearly mapped down to a size of 2, representing the input and output grids.[10] The log-likelihood is given by the crossentropy, with sum reduction across all the grids.

For grids of non-constant shape, the $[\text{example}, \text{height}, \text{channel}]$ and $[\text{example}, \text{width}, \text{channel}]$ tensors are used to create distributions over possible contiguous rectangular slices of each grid of colors,

---

[9]We also multiply the results by coefficients depending on the angle: 1 for 0 degrees and 180 degrees, 0.2 for 45 degrees and 135 degrees, and 0.4 for 90 degrees.

[10]The linear map is initialized to be identical for both the input and output grid, but isn't fixed this way during learning. Sometimes this empirically helps with problems of inconsistent input vs output grid shapes. The bias on this linear map is multiplied by 100 before usage, otherwise it doesn't seem to be learned fast enough empirically. This isn't done for the shape tensors described by the following paragraph though.

as shown in Figure 11. Again, the channel dimension is mapped down to a size of 2 for input and output grids. For every grid, we have a vector of size [width] and a vector of size [height]. The log likelihood of every slice of the vector is taken to be the sum of the values within the slice, minus the values outside the slice. The log likelihoods for all the possible slices are then normalized to have total probability one, and the colors for every slice are given by the color logits defined in the previous paragraph.

With the puzzle distribution now defined, we can now evaluate the log-likelihood of the observed target puzzle, to use as the reconstruction error.[11]

# E  OTHER ARCHITECTURAL DETAILS

## E.1  RULES FOR LEGAL MULTITENSORS

1. At least one non-example dimension must be included. Examples are not special for any reason not having to do with colors, directions, rows, and columns.

2. If the width or height dimension is included, the example dimension should also be included. Positions are intrinsic to grids, which are indexed by the example dimension. Without a grid it doesn't make as much sense to talk about positions.

## E.2  WEIGHT TYING FOR REFLECTION/ROTATION SYMMETRY

When applying a different linear layer to every tensor in a multitensor, we have a linear layer for tensors having a width but not height dimension, and another linear layer for tensors having a height but not width dimension. Whenever this is the case, we tie the weights together in order to preserve the whole network's equivariance to diagonal reflections and 90 degree rotations, which swap the width and height dimensions.

The softmax layer is not completely symmetrized because different indices of the output correspond to different combinations of dimension to softmax over. Tying the weights properly would be a bit complicated and time consuming for the performance improvement we expect, so we did not do this.

## E.3  TRAINING/INITIALIZATION

We train for 2000 iterations using Adam, with learning rate 0.01, $\beta_1$ of 0.5, and $\beta_2$ of 0.9. Weights are essentially all initialized with Xavier normal initialization.

# F  PREPROCESSING

## F.1  OUTPUT SHAPE DETERMINATION

The raw data consists of grids of various shapes, while the neural network operates on grids of constant shape. Most of the preprocessing that we do is aimed towards this shape inconsistency problem.

Before doing any training, we determine whether the given ARC-AGI puzzle follows three possible shape consistency rules:

1. The outputs in a given ARC-AGI puzzle are always the same shape as corresponding inputs.

2. All the inputs in the given ARC-AGI puzzle are the same shape.

3. All the outputs in the given ARC-AGI puzzle are the same shape.

---

[11]There are multiple slices of the same shape that result in the correct puzzle to be decoded. We sum together the probabilities of getting any of the slices by applying a logsumexp to the log probabilities. But, we found empirically that training prematurely collapses onto one particular slice. So, we pre-multiply and post-divide the log probabilities by a coefficient when applying the logsumexp. The coefficient starts at 0.1 and increases exponentially to 1 over the first 100 iterations of training. We also pre-multiply the masks by the square of this coefficient as well, to ensure they are not able to strongly concentrate on one slice too early in training.

Based on rules 1 and 3, we try to predict the shape of held-out outputs, prioritizing rule 1 over rule 3. If either rule holds, we force the postprocessing step to only consider the predicted shape by overwriting the masks produced by the linear head layer. If neither rule holds, we make a temporary prediction of the largest width and height out of the grids in the given ARC-AGI puzzle, and we allow the masks to predict shapes that are smaller than that.

The largest width and height that is given or predicted, are used as the size of the multitensor's width and height dimensions.

The predicted shapes are also used as masks when performing the multitensor communication, directional communication and directional cummax/shift layers. We did not apply masks for the other layers because of time constraints and because we do not believe it will provide for much of a performance improvement.[12]

### F.2 NUMBER OF COLORS

We notice that in almost all ARC-AGI puzzles, colors that are not present in the puzzle are not present in the true answers. Hence, any colors that do not appear in the puzzle are not given an index in the color dimension of the multitensor.

In addition, black is treated as a special color that is never included in the multitensor, since it normally represents the background in many puzzles. When performing color classification, a tensor of zeros is appended to the color dimension after applying the linear head, to represent logits for the black color.

## G POSTPROCESSING

Since the generated answer grid is stochastic from randomness in $z$, we save the answer grids throughout training, and roughly speaking, we choose the most frequently occuring one as our denoised final prediction. This is complicated by the variable shape grids present in some puzzles.

Generally, when we sample answers from the network by taking the logits of the [example, color, height, width, channel] tensor and argmaxxing over the color dimension, we find that the grids are noisy and will often have the wrong colors for several random pixels. We developed several methods for removing this noise:

1. Find the most commonly sampled answer.

2. Construct an exponential moving average of the output color logits before taking the softmax to produce probabilities. Also construct an exponential moving average of the masks.

3. Construct an exponential moving average of the output color probabilities after taking the softmax. Also construct an exponential moving average of the masks.

When applying these techniques, we always take the slice of highest probability given the mask, and then we take the colors of highest probability afterwards.

We explored several different rules for when to select which method, and arrived at a combination of 1 and 2 with a few modifications:

- At every iteration, count up the sampled answer, as well as the exponential moving average answer (decay = 0.97).

- If before 150 iterations of training, then downweight the answer by a factor of $e^{-10}$. (Effectively, don't count the answer.)

- If the answer is from the exponential moving average as opposed to the sample, then downweight the answer by a factor of $e^{-4}$.

- Downweight the answer by a factor of $e^{-10*\text{uncertainty}}$, where uncertainty is the average (across pixels) negative log probability assigned to the top color of every pixel.

---

[12]The two masks for the input and output are combined together to make one mask for use in these operations, since the channel dimension in these operations don't necessarily correspond to the input and output grids.

# H  EMPIRICALLY OBSERVED ABILITIES AND DISABILITIES OF COMPRESSARC

(b) CompressARC's solution to puzzle 28e73c20

(a) Puzzle 28e73c20

Figure 12: Puzzle 28e73c20, and CompressARC's solution to it.

A short list of abilities that **can** be performed by CompressARC includes:

- Assigning individual colors to individual procedures (see puzzle 0ca9ddb6)
- Infilling (see puzzle 0dfd9992)
- Cropping (see puzzle 1c786137)
- Connecting dots with lines, including 45 degree diagonal lines (see puzzle 1f876c06)
- Same color detection (see puzzle 1f876c06)
- Identifying pixel adjacencies (see puzzle 42a50994)
- Assigning individual colors to individual examples (see puzzle 3bd67248)
- Identifying parts of a shape (see puzzle 025d127b)
- Translation by short distances (see puzzle 025d127b)

We believe these abilities to be individually endowed by select layers in the architecture, which we designed specifically for the purpose of conferring those abilities to CompressARC.

A short list of abilities that **cannot** be performed by CompressARC includes:

- Assigning two colors to each other (see puzzle 0d3d703e)
- Repeating an operation in series many times (see puzzle 0a938d79)
- Counting/numbers (see puzzle ce9e57f2)
- Translation, rotation, reflections, rescaling, image duplication (see puzzles 0e206a2e, 5ad4f10b, and 2bcee788)
- Detecting topological properties such as connectivity (see puzzle 7b6016b9)
- Planning, simulating the behavior of an agent (see puzzle 2dd70a9a)
- Long range extensions of patterns (see puzzle 28e73c20 above)

# I  BASELINES

The U-Net baseline in Table 1 was created to observe the performance of a more standard approach when subject to the same constraints of CompressARC, namely the avoidance of any training before inference time, and the sole use of the test puzzle as training data during inference time.

The training algorithm for the baseline consists of feeding each input grid into the U-Net and using the U-Net output to classify the pixel color of the output grid. Puzzles where the input grid and output grid did not match shape were skipped and assumed to receive a score of zero. The most common two output grids occurring in the second half of the 10000 steps of training were used as the two solution guesses.

We did not change the width or height of the grids in order to fit the ARC-AGI grids into the U-Net. The U-Net's BatchNorm was replaced with a GroupNorm, and the middle pooling/upsampling layers were skipped if the activation grids were too small to be pooled anymore.

We experimented with applying a random augmentation transformation to the input grids and reversing the transformation on the output before computing the loss and/or ensembling predictions, but we discarded this idea due to worse performance on the training set (2.5% without augmentations, 1% with augmentations).

# J  PUZZLE SOLVE ACCURACY TABLES

See Tables 4 and 5 for numerically reported puzzle solve accuracies on the whole dataset.

Table 4: CompressARC's puzzle solve accuracy on the training set as a function of the number of steps of inference time learning it is given, for various numbers of allowed guesses (pass@n). The official benchmark is reported with 2 allowed guesses, which is why we report 20% on the evaluation set. Total training set solve time is reported for an NVIDIA RTX 4070 GPU by solving one puzzle at a time in a sequence.

| Training Iteration | Time | Pass@1 | Pass@2 | Pass@5 | Pass@10 | Pass@100 | Pass@1000 |
|---|---|---|---|---|---|---|---|
| 100 | 6 h | 1.00% | 2.25% | 3.50% | 4.75% | 6.75% | 6.75% |
| 200 | 13 h | 11.50% | 14.25% | 16.50% | 18.25% | 23.25% | 23.50% |
| 300 | 19 h | 18.50% | 21.25% | 23.50% | 26.75% | 31.50% | 32.50% |
| 400 | 26 h | 21.00% | 25.00% | 28.75% | 31.00% | 36.00% | 37.50% |
| 500 | 32 h | 23.00% | 27.50% | 31.50% | 33.50% | 39.25% | 40.75% |
| 750 | 49 h | 28.00% | 30.50% | 34.00% | 36.25% | 42.75% | 44.50% |
| 1000 | 65 h | 28.00% | 31.75% | 35.50% | 37.75% | 43.75% | 46.50% |
| 1250 | 81 h | 29.00% | 32.25% | 37.00% | 39.25% | 45.50% | 49.25% |
| 1500 | 97 h | 29.50% | 33.00% | 38.25% | 40.75% | 46.75% | 51.75% |
| 2000 | 130 h | 30.25% | 34.75% | 38.25% | 41.50% | 48.50% | 52.75% |

# K  HOW TO IMPROVE OUR WORK

At the time of release of CompressARC, there were several ideas which we thought of trying or attempted at some point, but didn't manage to get working for one reason or another. Some ideas we still believe in, but didn't use, are listed below.

## K.1  JOINT COMPRESSION VIA WEIGHT SHARING BETWEEN PUZZLES

Template Algorithm 1 includes a hard-coded value of $\theta$ for every single puzzle. We might be able to further shorten the template program length by sharing a single $\theta$ between all the puzzles, knowing that Occam's razor says a shorter program corresponds to more correct puzzle solutions. Algorithm 2 would have to be changed accordingly.

To implement this, we would most likely explore strategies like:

Table 5: CompressARC's puzzle solve accuracy on the evaluation set, reported the same way as in Table 4.

| Training Iteration | Time | Pass@1 | Pass@2 | Pass@5 | Pass@10 | Pass@100 | Pass@1000 |
|---|---|---|---|---|---|---|---|
| 100 | 7 h | 0.75% | 1.25% | 2.25% | 2.50% | 3.00% | 3.00% |
| 200 | 14 h | 5.00% | 6.00% | 7.00% | 7.75% | 12.00% | 12.25% |
| 300 | 21 h | 10.00% | 10.75% | 12.25% | 13.25% | 15.50% | 16.25% |
| 400 | 28 h | 11.75% | 13.75% | 16.00% | 17.00% | 19.75% | 20.00% |
| 500 | 34 h | 13.50% | 15.00% | 17.75% | 19.25% | 20.50% | 21.50% |
| 750 | 52 h | 15.50% | 17.75% | 19.75% | 21.50% | 22.75% | 25.50% |
| 1000 | 69 h | 16.75% | 19.25% | 21.75% | 23.00% | 26.00% | 28.75% |
| 1250 | 86 h | 17.00% | 20.75% | 23.00% | 24.50% | 28.25% | 30.75% |
| 1500 | 103 h | 18.25% | 21.50% | 24.25% | 25.50% | 29.50% | 31.75% |
| 2000 | 138 h | 18.50% | 20.00% | 24.25% | 26.00% | 31.25% | 33.75% |

- Using the same network weights for all puzzles, and training for puzzles in parallel. Each puzzle gets assigned some perturbation to the weights, that is constrained in some way, e.g., LORA (Hu et al., 2021).

- Learning a "puzzle embedding" for every puzzle that is a high dimensional vector (more than 16 dim, less than 256 dim), and learning a linear mapping from puzzle embeddings to weights for our network. This mapping serves as a basic hypernetwork, i.e., a neural network that outputs weights for another neural network (Chauhan et al., 2024).

Unfortunately, testing this would require changing CompressARC (Algorithm 3) to run all puzzles in parallel rather than one at a time in series. This would slow down the research iteration process, which is why we did not explore this option.

### K.2 Convolution-like Layers for Shape Copying Tasks

This improvement is more ARC-AGI-specific and may have less to do with AGI in our view. Many ARC-AGI-1 puzzles can be seen to involve copying shapes from one place to another, and our network has no inductive biases for such an operation. An operation which is capable of copying shapes onto multiple locations is the convolution. With one grid storing the shape and another with pixels activated at locations to copy to, convolving the two grids will produce another grid with the shape copied to the designated locations.

There are several issues with introducing a convolutional operation for the network to use. Ideally, we would read two grids via projection from the residual stream, convolve them, and write it back in via another projection, with norms in the right places and such. Ignoring the fact that the grid size changes during convolution (can be solved with two parallel networks using different grid sizes), the bigger problem is that convolutions tend to amplify noise in the grids much more than the sparse signals, so their inductive bias is not good for shape copying. We can try to apply a softmax to one or both of the grids to reduce the noise (and to draw an interesting connection to attention), but we didn't find any success.

The last idea that we were tried before discarding the idea was to modify the functional form of the convolution:

$$(f * g)(x) = \sum_y f(x - y)g(y)$$

to a tropical convolution (Fan et al., 2021), which we found to work well on toy puzzles, but not well enough for ARC-AGI-1 training puzzles (which is why we discarded this idea):

$$(f * g)(x) = \max_y f(x - y) + g(y)$$

Convolutions, when repeated with some grids flipped by 180 degrees, tend to create high activations at the center pixel, so sometimes it is important to zero out the center pixel to preserve the signal.

### K.3 KL FLOOR FOR POSTERIOR COLLAPSE

We noticed during testing that crucial posterior tensors whose KL fell to zero during learning would never make a recovery and play their role in the encoding, just as in the phenomenon of mode collapse in variational autoencoders (van den Oord et al., 2018). We believe that the KL divergence may upper bound the information content of the gradient training signal for parts of the network that process the encoded information. Thus, when a tensor in $z$ falls to zero KL, the network stops learning to use its encoded information, and the KL is no longer incentivized to recover. If we artificially hold the KL above zero for an extended period of training, then the network may learn to make use the tensor's information, incentivizing the KL to stay above zero when released again.

We implemented a mechanism to keep the KL above a minimum threshold so that the network always learns to use that information, but we do not believe the network learns fast enough for this to be useful, as we have never seen a tensor recover before. Therefore, it might be useful to explore different ways to schedule this KL floor to start high and decay to zero, to allow learning when the KL is forced to be high, and to leave the KL unaffected later on in learning. This might cause training results to be more consistent across runs.

### K.4 REGULARIZATION

In template Algorithm 1, we do not code-golf $\theta$ to reduce the number of bits it takes up. If we were to code-golf $\theta$ as well, this would produce an extra KL term to the loss in CompressARC (Algorithm 3), and the KL term would simplify to an L2 regularization on $\theta$ under certain reasonable limits. It is somewhat reckless for us to neglect code-golfing $\theta$ in our work due to the sheer number of bits $\theta$ contributes, and making this change may improve our results.

## L ADDITIONAL DETAILS ABOUT THE ARC-AGI BENCHMARK

**Hidden Rules:** For every puzzle, there is a hidden rule that maps each input grid to each output grid. There are 400 training puzzles and they are easier to solve than the 400 evaluation puzzles. The training set is intended to help teach your system the following general themes which underlie the hidden rules in the evaluation set:

- **Objectness**: Objects persist and cannot appear or disappear without reason. Objects can interact or not depending on the circumstances.
- **Goal-directedness**: Objects can be animate or inanimate. Some objects are "agents" - they have intentions and they pursue goals.
- **Numbers & counting**: Objects can be counted or sorted by their shape, appearance, or movement using basic mathematics like addition, subtraction, and comparison.
- **Basic geometry & topology**: Objects can be shapes like rectangles, triangles, and circles which can be mirrored, rotated, translated, deformed, combined, repeated, etc. Differences in distances can be detected.

The puzzles are designed so that **humans can reasonably find the answer, but machines should have more difficulty**. The average human can solve 76.2% of the training set, and a human expert can solve 98.5% (LeGris et al., 2024).

**Scoring:** You are given some number of examples of input-to-output mappings, and you get **two guesses** to guess the output grid(s) for a given input grid, without being told the hidden rule. One guess consists of guessing the width and height of the output grid(s), as well as all the pixel colors within, and if each of these is correct, then that guess succeeds. If either guess succeeds, then you score 1 for that puzzle, else you score 0. Some puzzles have more than one input/output pair that you have to guess, in which case the score for that puzzle may be in between.

**Scoring Environment:** The competitions launched by the ARC Prize Foundation have been restricted to 12 hours of compute per solution submission, in a constrained environment with no internet access.

This is where a hidden semi-private evaluation set is used to score solutions. The scores we report are on the public evaluation set, which is of the same difficulty as the semi-private evaluation set, which we had no access to when we performed this work.

# M  ADDITIONAL CASE STUDIES

Below, we show two additional puzzles and a dissection of CompressARC's solution to them.

## M.1  CASE STUDY: BOUNDING BOX

Puzzle 6d75e8bb is part of the training split, see Figure 13.

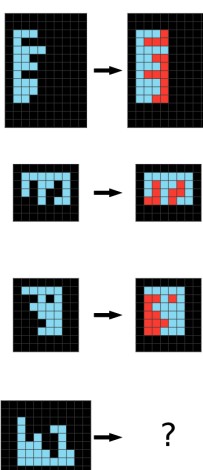

Figure 13: Bounding Box: Puzzle 6d75e8bb from the training split.

### M.1.1  WATCHING THE NETWORK LEARN: BOUNDING BOX

**Human Solution:** We first realize that the input is red and black, and the output is also red and black, but some of the black pixels are replaced by light blue pixels. We see that the red shape remains unaffected. We notice that the light blue box surrounds the red shape, and finally that it is the smallest possible surrounding box that contains the red shape. At this point, we copy the input over to the answer grid, then we figure out the horizontal and vertical extent of the red shape, and color all of the non-red pixels within that extent as light blue.

**CompressARC Solution: See Table 6**

### M.1.2  SOLUTION ANALYSIS: BOUNDING BOX

Figure 14 shows the amount of contained information in every tensor within $z$.

All the tensors in $z$ fall to zero information content during training, except for three tensors. From 600-1000 steps, we see the (example, height, width, channel) tensor suffer a massive drop in information content, with no change in the outputted answer. We believe it was being used to identify the light blue pixels in the input, but this information then got memorized by the nonlinear portions of the network, using the (example, height, channel) and (example, width, channel) as positional encodings.

Figure 15 shows the average output of the decoding layer for these tensors to see what information is stored there.

## M.2  CASE STUDY: CENTER CROSS

Puzzle 41e4d17e is part of the training split, see Figure 16a.

Table 6: CompressARC learning the solution for Bounding Box, over time.

| Learning steps | What is CompressARC doing? | Sampled solution guess |
|---|---|---|
| 50 | The average of sampled outputs shows that light blue pixels in the input are generally preserved in the output. However, black pixels in the input are haphazardly and randomly colored light blue and red. CompressARC does not seem to know that the colored input/output pixels lie within some kind of bounding box, or that the bounding box is the same for the input and output grids. |  |
| 100 | The average of sampled outputs shows red pixels confined to an imaginary rectangle surrounding the light blue pixels. CompressARC seems to have perceived that other examples use a common bounding box for the input and output pixels, but is not completely sure about where the boundary lies and what colors go inside the box in the output. Nevertheless, guess 2 (the second most frequently sampled output) shows that the correct answer is already being sampled quite often now. |  |
| 150 | The average of sampled outputs shows almost all of the pixels in the imaginary bounding box to be colored red. CompressARC has figured out the answer, and further training only refines the answer. |  |

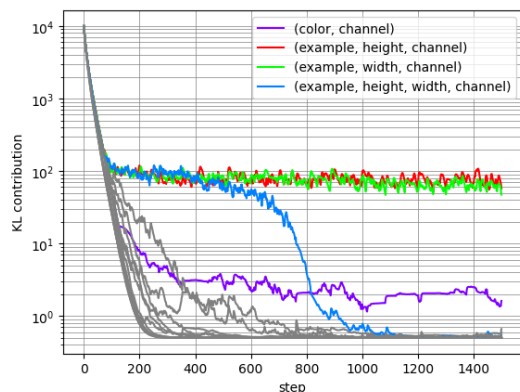

Figure 14: Breaking down the KL loss during training into contributions from each individual shaped tensor in the multitensor $z$.

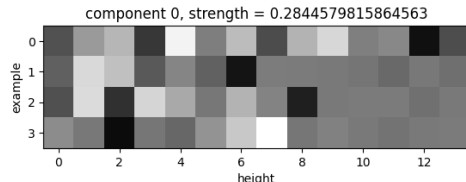

(a) (**example**, **height**, **channel**) **tensor.** The first principal component is 771 times stronger than the second principal component. **A brighter pixel indicates a row with more light blue pixels.** It is unclear how CompressARC knows where the borders of the bounding box are.

(b) (**example**, **width**, **channel**) **tensor.** The first principal component is 550 times stronger than the second principal component. **A darker pixel indicates a column with more light blue pixels.** It is unclear how CompressARC knows where the borders of the bounding box are.

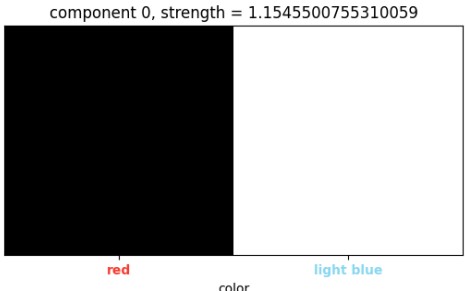

(c) (**color**, **channel**) **tensor.** This tensor serves to distinguish the roles of the two colors apart.

Figure 15: Breaking down the loss components during training tells us where and how CompressARC prefers to store information relevant to solving a puzzle.

**Human Solution:** We first notice that the input consists of blue "bubble" shapes (really they are just squares, but the fact that they're blue reminds us of bubbles) on a light blue background and the output has the same. But in the output, there are now magenta rays emanating from the center of each bubble. We copy the input over to the answer grid, and then draw magenta rays starting from the center of each bubble out to the edge in every cardinal direction. At this point, we submit our answer and find that it is wrong, and we notice that in the given demonstrations, the blue bubble color is drawn on top of the magenta rays, and we have drawn the rays on top of the bubbles instead. So, we pick up the blue color and correct each point where a ray pierces a bubble, back to blue.

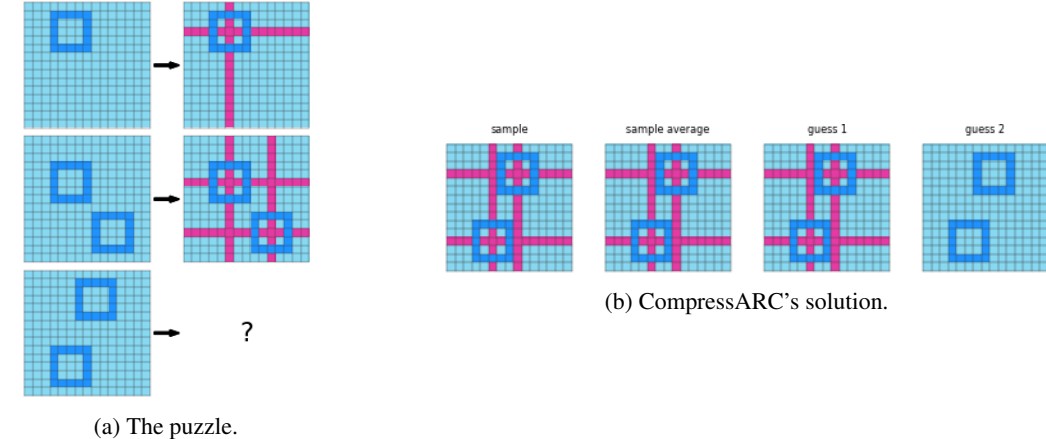

(b) CompressARC's solution.

(a) The puzzle.

Figure 16: Center Cross: Puzzle 41e4d17e from the training split.

**CompressARC Solution:** We don't show CompressARC's solution evolving over time because we think it is uninteresting; instead will describe. We don't see much change in CompressARC's answer over time during learning. It starts by copying over the input grid, and at some point, magenta rows and columns start to appear, and they slowly settle on the correct positions. At no point does CompressARC mistakenly draw the rays on top of the bubbles; it has always had the order correct.

### M.2.1 SOLUTION ANALYSIS: CENTER CROSS

Figure 17 shows another plot of the amount of information in every tensor in $z$. The only surviving tensors are the $(\text{color}, \text{channel})$ and $(\text{example}, \text{height}, \text{width}, \text{channel})$ tensors, which are interpreted in Figure 18.

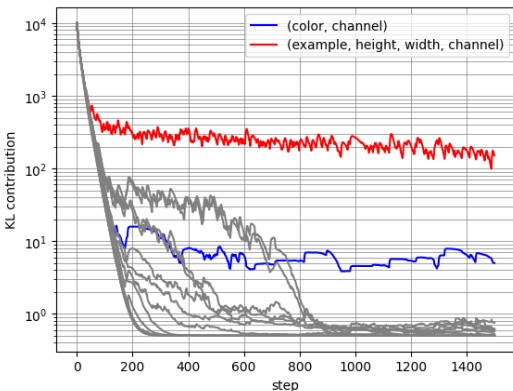

Figure 17: Breaking down the KL loss during training into contributions from each individual shaped tensor in the multitensor $z$.

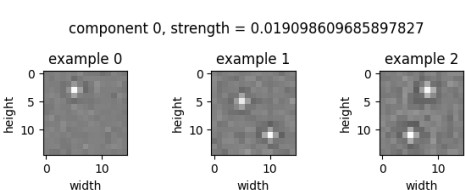

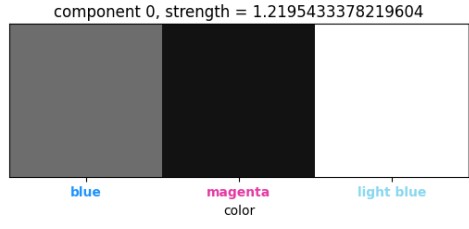

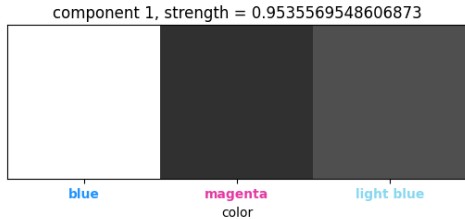

(a) (**example**, **height**, **width**, **channel**) **tensor.** The top principal component is 2496 times stronger than the second principal component. **This tensor codes for the centers of the bubbles.** In the KL contribution plot, we can see that the information content of this tensor is decreasing over time. Likely, CompressARC is in the process of eliminating the plus shaped representation, and replacing it with a pixel instead, which takes fewer bits.

(b) (**color**, **channel**) **tensor.** This tensor just serves to distinguish the individual roles of the colors in the puzzle.

Figure 18: Breaking down the loss components during training tells us where and how CompressARC prefers to store information relevant to solving a puzzle.

# N LIST OF MENTIONED ARC-AGI-1 PUZZLES

See Table 7 below.

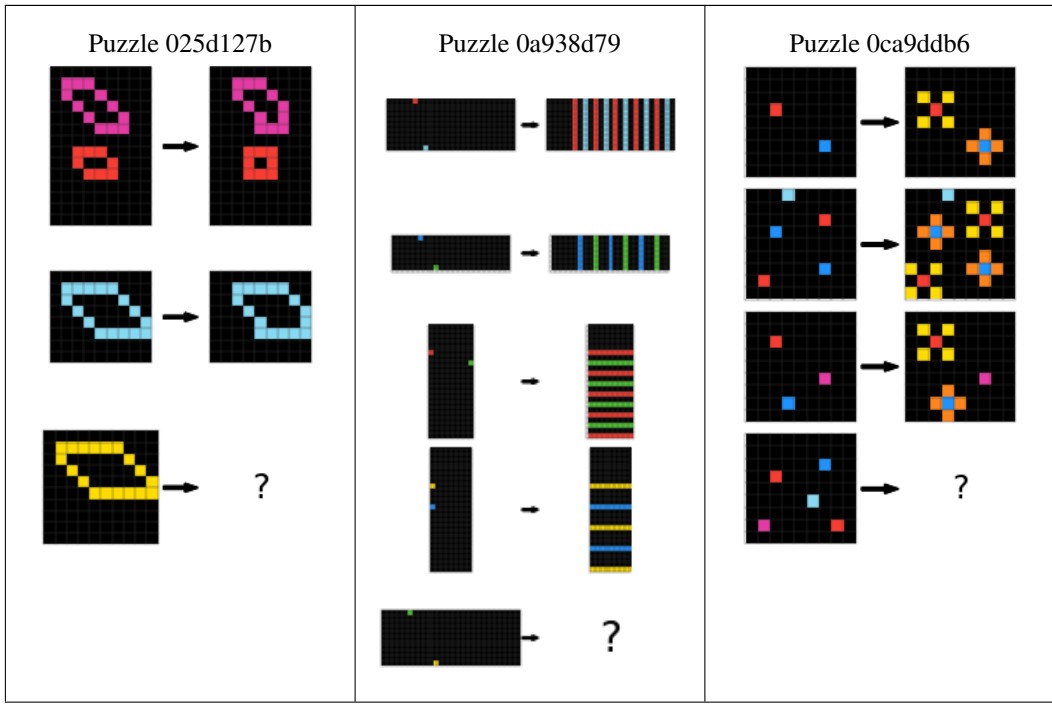

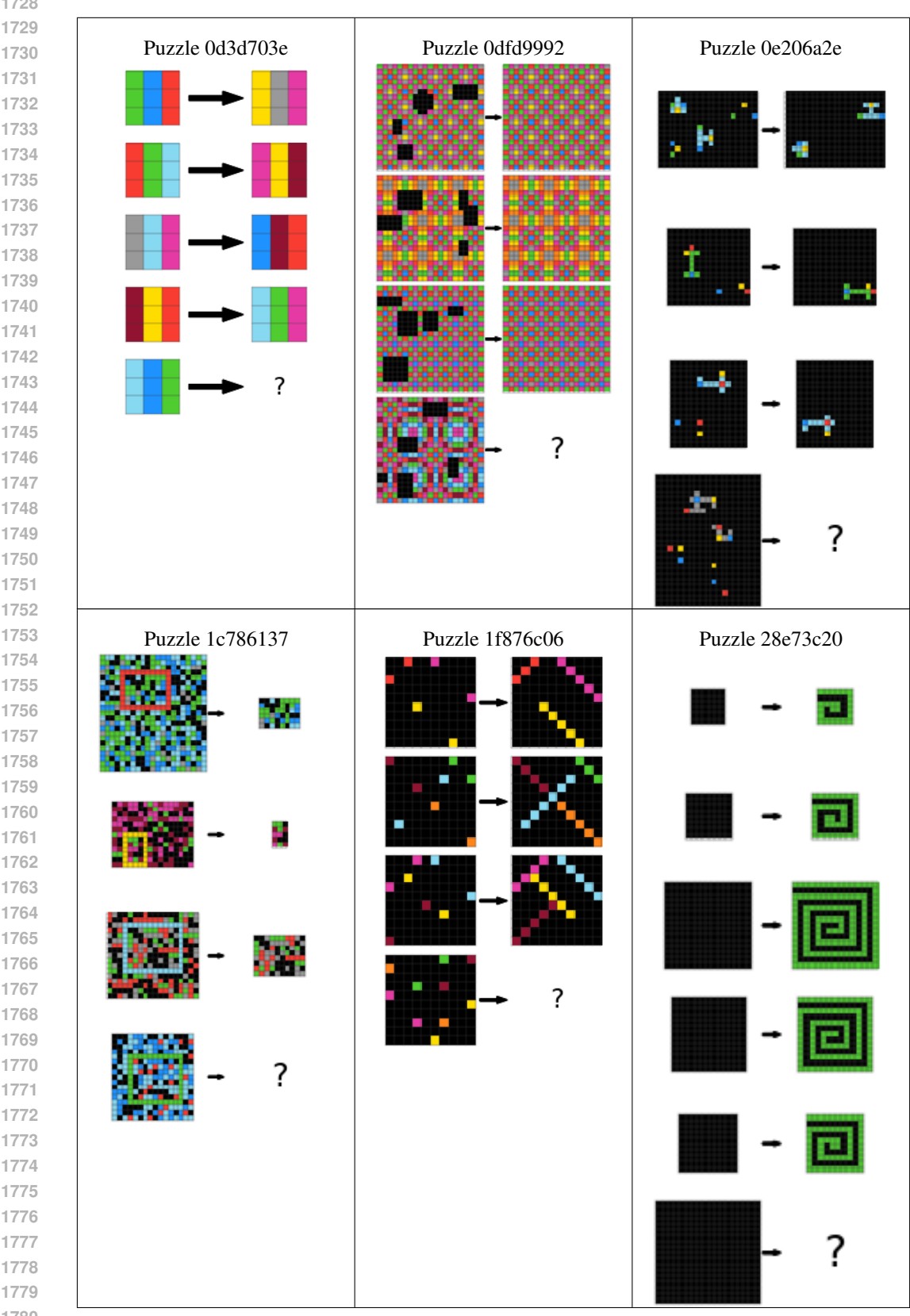

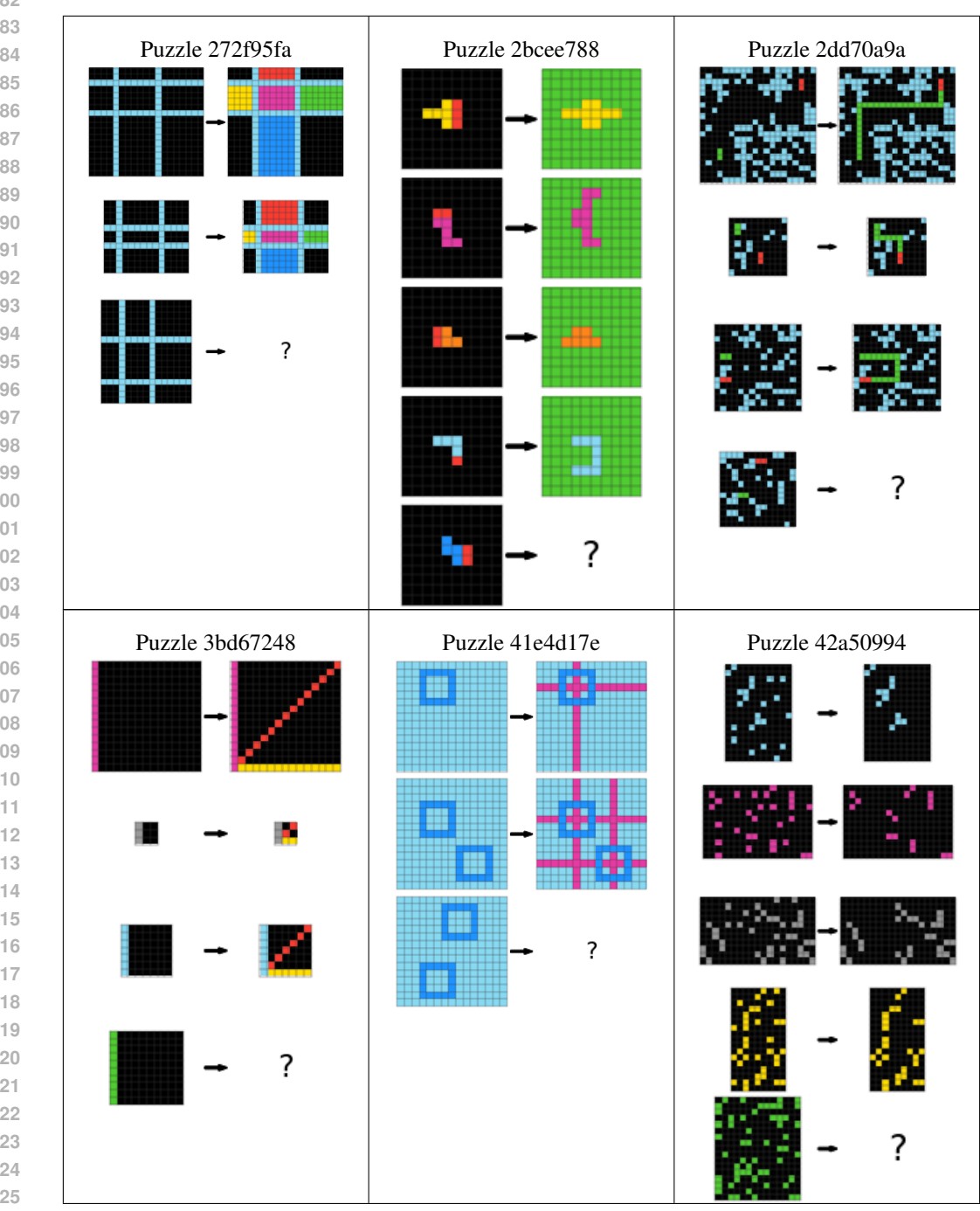

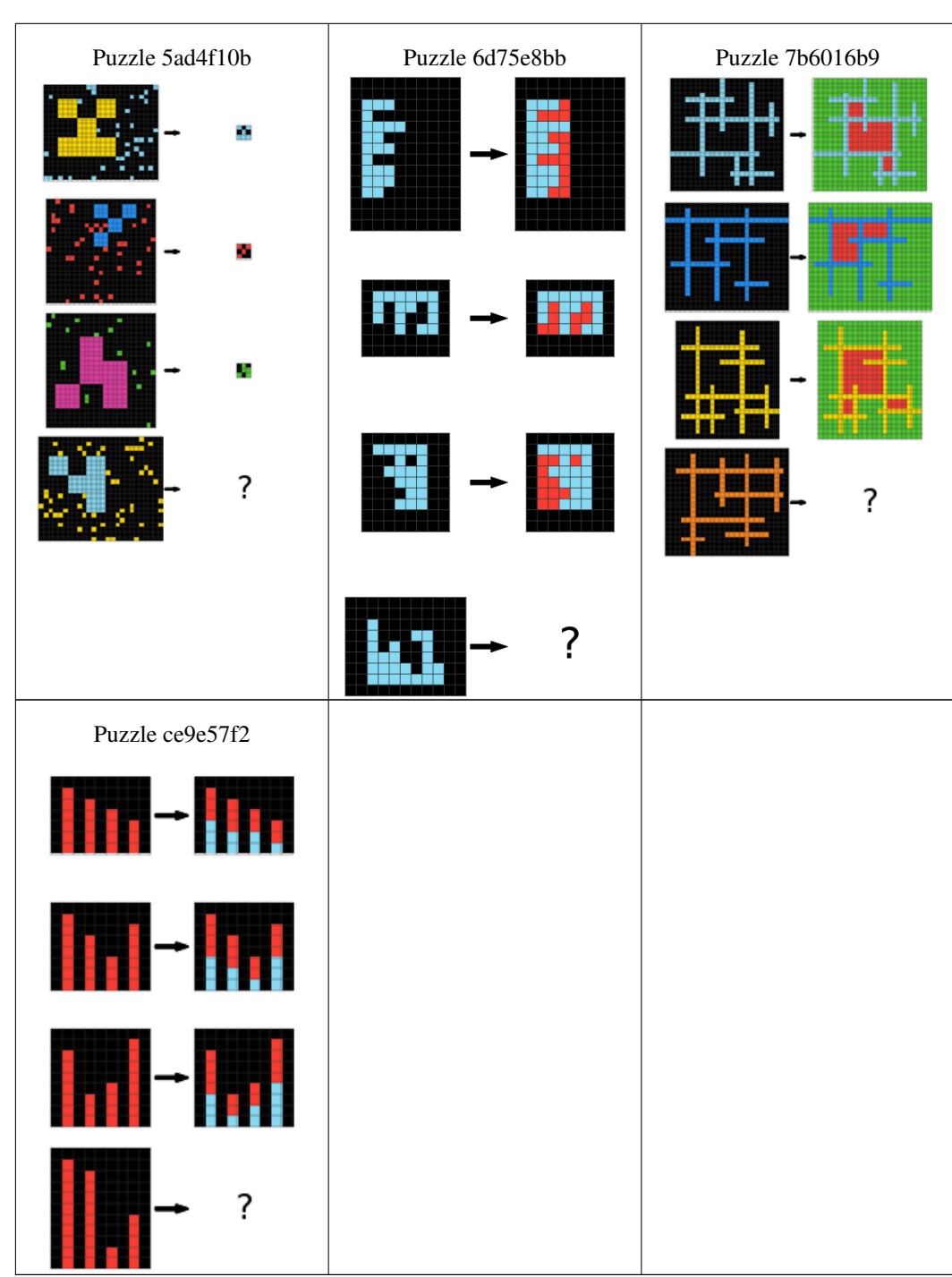

Table 7: List of Mentioned ARC-AGI=1 Puzzles. All these puzzles are part of the training split.

## O CODE

Code for this project is provided in the supplemental materials.

