# OpenReview forum: "ARC-AGI Without Pretraining"
_ICLR.cc/2026/Conference — Submitted to ICLR 2026_

### Official Review · Reviewer_zdRF · 2025-10-30

**Soundness:** 3
**Presentation:** 2
**Contribution:** 3
**Rating:** 6
**Confidence:** 3

**Summary:**

This work introduces a method for solving ARC-AGI-1 evaluation puzzles, based on minimum description length approach (i.e., the so-called code golf). It is worthy of noting that this work does not requite any pretraining, and only trains a rather small neural network during inference time, showing potentials for efficiency and the rid of massive LLMs training.

**Strengths:**

As shown in the summary:
1. A novel inference-time perspective for ARC-AGI, as the pertaining is omitted.
2. Rich discussions on the proposed schemes in the solving process and explanations.
3. Clarity for pros and cons in the work.

**Weaknesses:**

1. The proposed pipeline may lack formal theoretical analyses for the mathematical  fundaments (which though I think applies to other related methods in this field).
2. Is it possible to apply MDL to other tasks? How and why is possible/promising?
3. It seems that the running time for (total iterations and per iteration) remain possible to be greatly improved. How do the author explain such cost/efficiency and what can be done for improvement?
4. Despite its focus on the pretraining-free advanatage, it would still be nice to have more baselines(related work) comparisons and discussions on the main context.

**Questions:**

See the weakness.

---

> ### Author Response · Authors · 2025-11-26
>
> We thank the reviewer for taking the time to evaluate our submission and providing feedback to help us improve our work. We have made changes in the paper to reflect any changes we offer in this rebuttal; they are visible in the updated version of the paper.
>
> > **The proposed pipeline may lack formal theoretical analyses for the mathematical fundaments (which though I think applies to other related methods in this field).**
>
> When it comes to the theoretically optimally code-golfed solution and it's properties in selecting "good" solutions, there is an existing body of literature, which we cite but do not explain in depth. We will add more citations to this field in the revised version (lines 197-205), and expand our analysis in Appendix A. For our candidate program and it's optimization, we also refer to existing literature on relative entropy coding (REC), which irons out meticulous details regarding the definition of "imitating a sampling distribution", seed length approximation bounds, and computation speed. In our revision, we will include a note at the end of Section 3 to refer readers to this literature for those specific details.
>
> When it comes to why the program length (i.e., Kolmogorov complexity) is good to use as "Occam's complexity" for Occam's razor, we indeed offer no explanation except that Kolmogorov complexity has conceptually appealing mathematical properties [2].
>
> > **Is it possible to apply MDL to other tasks? How and why is possible/promising?**
>
> It is indeed possible to apply MDL to recreate the equivalent of CompressARC in a wide variety of domains, except that when you perform an equivalent derivation of Algorithms 1, 2, and 3 on another domain, it can be hard to avoid recovering already-known, successful algorithms at the end of the process. For example, for discrete sequential data, you can pick Algo 1 to be:
>
> ```
> 1. Define a function f which maps from a space X^n to a probability distribution over space X, that works for any n.
> 2. Set seed = <hardcoded value>.
> 3. Set sequence as empty list.
> 4. For index from 1 to length:
> 5.     distribution = f(sequence)
> 6.     item = sample(distribution, seed)
> 7.     Append item to sequence.
> 8. Return sequence
> ```
>
> You would then find that to code-golf some given sequence (where some indices are not known), the expected length of the seed you would have to pick is the negative log probability of sampling the known parts of the given sequence (i.e., the crossentropy) when you unroll the function f. Then minimizing the program length boils down to minimizing the seed length, which with some math, amounts to learning a function f to minimize the crossentropy. Voila, we have discovered autoregressive sequence modeling with a restriction for the model to be causal. Sounds familiar, except note that there is no restriction that the model cannot be trained during inference time here. There is similar one that reproduces diffusion models [1], which requires a bit more work on Algorithm 3, and so on. We do not go into these details in this paper because it might be worth a whole other paper, but hopefully you can see that MDL is already present in many existing successful paradigms, and may be convinced that it can be used to derive new ones.
>
> > **It seems that the running time for (total iterations and per iteration) remain possible to be greatly improved. How do the author explain such cost/efficiency and what can be done for improvement?**
>
> In general, the code in it's current state is not optimized for running time performance, since we make use of a many custom operations that we have not done any engineering to optimize. There is no structural barrier preventing a faster model, and GPU optimizations (JIT compilation, specialized kernels) can alleviate this issue, as with any kind of model. We will include these details in the Discussion in the revised version. The supplementary materials also contains code for parallelizing to solve multiple puzzles at once.
>
> > **Despite its focus on the pretraining-free advantage, it would still be nice to have more baselines(related work) comparisons and discussions on the main context.**
>
> Multiple reviewers have made similar comments; we have decided to include an inference-time-trained U-Net as a baselines for comparison, and we will add it in Table 1 in the revised version, also including comparisons from related work (which we moved to the top of the Appendix) to give context.
>
> [1] Ho, J., Jain, A., & Abbeel, P. (2020). Denoising diffusion probabilistic models. Advances in neural information processing systems, 33, 6840-6851.
>
> [2] Kolmogorov, A. N. (1998). On tables of random numbers. Theoretical Computer Science, 207(2), 387–395. doi:10.1016/S0304-3975(98)00075-9

---

> > ### Comment · Reviewer_zdRF · 2025-11-28
> > **Reply to Rebuttal**
> >
> > I would like to thank the authors for the replies. Regarding the rebuttal and the current manuscript, weakness #3 has not been addressed, as it takes efforts to further enhance efficiency; I acknowledge that others have been made clearer and partially resolved (though not comprehensively), considering limited time and the current version. I will maintain my scores.

---

### Official Review · Reviewer_Y2bJ · 2025-11-01

**Soundness:** 2
**Presentation:** 2
**Contribution:** 1
**Rating:** 2
**Confidence:** 5

**Summary:**

This paper introduces a lightweight CompressARC to solve puzzles from the ARC-AGI-1 benchmark without any pretraining. By applying the Minimum Description Length (MDL) principle during inference, the model learns purely from the target puzzle itself, without using any training set. The results suggest MDL as a data-efficient solution for solving puzzles.

**Strengths:**

1.	The research addresses a compelling and timely question on data-efficient methods for solving ARC-AGI puzzles, offering a novel alternative to large-scale pretraining.
2.	The manuscript and supplementary materials provide comprehensive implementation details, ensuring reproducibility and methodological clarity.
3.	The experimental results are supported by in-depth analysis of the proposed method.

**Weaknesses:**

1.	The emphasis on "no pretraining" and single-puzzle focus does not sufficiently establish methodological novelty; it remains unclear whether performance stems from genuine innovation or inherent dataset shortcuts.
2.	Claims of novelty are inadequately supported by architectural or algorithmic innovation, as the model relies on established components without clear differentiation.
3.	Section 3 lacks intuitive motivation, proceeding directly into technical details without conceptual justification, hindering reader comprehension.
4.	The three core algorithms appear to lack substantive novelty, with insufficient emphasis on what constitutes a technical advance.
5.	The network architecture is relatively conventional, predominantly building on widely adopted components.
6.	Comparative evaluation with state-of-the-art methods is absent, limiting validation of claimed advantages.
7.	Generalization claims are not fully convincing, as the method may exploit dataset-specific shortcuts rather than learning generalizable reasoning, limiting applicability to tasks like RAVENs.

**Questions:**

1.	What specifically constitutes the novel component(s) in the proposed architecture or algorithm?
2.	Appendix K offers supplementary dataset details but lacks illustrative puzzle examples such as the structure of the question panel, the format of a correct solution, and the criteria for matching a proposed answer to the ground truth. Including such examples would significantly improve readers' understanding of the task and the proposed method's problem-solving process.
3.	During inference-time learning, are solutions to training/test samples used to fine-tune the model parameters?
4.	How are "steps" and "attempts" formally defined? What operations occur per step, and what constitutes an attempt?
5.	How does the method compare to existing few-shot or zero-shot learning approaches on comparable tasks?

---

> ### Author Response · Authors · 2025-11-26
>
> We thank the reviewer for taking the time to evaluate our submission and providing feedback to help us improve our work. We have made many changes to the paper to address each concern, which will be visible in the revised version which we have uploaded.
>
> > **The emphasis on "no pretraining" and single-puzzle focus does not sufficiently establish methodological novelty; it remains unclear whether performance stems from genuine innovation or inherent dataset shortcuts.**
>
> > **Generalization claims are not fully convincing, as the method may exploit dataset-specific shortcuts rather than learning generalizable reasoning, limiting applicability to tasks like RAVENs.**
>
> The prevailing perspective in machine learning has been that large amounts of data are required to produce intelligence. Since our method brings intelligence to an extremely data-efficient regime, we consider the single-puzzle test-time-only training focus to be novel. Can you provide some references showing similar work elsewhere?
>
> We do recognize the reviewer's concern,  and we updated our paper to include a standard supervised baseline where we use a U-Net [1] in the same pretraining-free inference-time-only setting but without our architecture or training, to try to few-shot learn the input-output mapping. The U-Net achieves a 0.75% score versus CompressARC's 20%, indicating that switching from the few-shot supervised approach to the novel compressive components is essential to improved performance.
>
> > **Comparative evaluation with state-of-the-art methods is absent, limiting validation of claimed advantages.**
>
> We realize that our evaluation of SOTA LLM methods is difficult to find because it is currently lumped into the ARC-AGI background (Appendix K, now Appendix L in new revision) instead of the Related Work (Appendix I, now Appendix A.4 in new revision). We will move this evaluation over to the Related Work. We will also include more comparison to these methods in a new Table 1, and highlight there that they require internet-scale pretraining to function properly where our method does not.
>
> > **Claims of novelty are inadequately supported by architectural or algorithmic innovation, as the model relies on established components without clear differentiation.**
>
> > **The network architecture is relatively conventional, predominantly building on widely adopted components.**
>
> > **The three core algorithms appear to lack substantive novelty, with insufficient emphasis on what constitutes a technical advance.**
>
> > **What specifically constitutes the novel component(s) in the proposed architecture or algorithm?**
>
> The three code-golf algorithms and multitensor networks are completely novel, since they arise from the code-golfing formulation, which nobody else has applied to ARC-AGI using neural networks. Where have you seen these components before and in what way are they already well-established, can you provide some references? We will make it clearer in the revised paper that we are the first to successfully solve many ARC-AGI puzzles by training a neural network solely during inference time.
>
> > **Section 3 lacks intuitive motivation, proceeding directly into technical details without conceptual justification, hindering reader comprehension.**
>
> In the introduction, we explain that our motivation for approaching the problem as a code-golfing problem is because Occam's razor instructs that this will lead to reproducing the correct solution. We will make it clearer at the top of Section 3 (explaining the methods) of the revised paper that this is the case.
>
> > **Appendix K offers supplementary dataset details but lacks illustrative puzzle examples such as the structure of the question panel, the format of a correct solution, and the criteria for matching a proposed answer to the ground truth. Including such examples would significantly improve readers' understanding of the task and the proposed method's problem-solving process.**
>
> Figure 12, in the middle of Appendix K, directly shows three example puzzles in the form of the question panel. The format of the solution is a grid, just like the grids shown in the Figure. The scoring criterion is if the length and height are the same, and every single pixel matches color. We will include this detail in the revised version, and move Figure 12 up to the main text on ARC-AGI background.
>
> > **During inference-time learning, are solutions to training/test samples used to fine-tune the model parameters?**
>
> Inference-time learning only occurs using one puzzle, the test puzzle. The solutions to the demonstration examples in that test puzzle are indeed used as they are part of the puzzle ($P$ is used during learning on line 10 of Algorithm 3).

---

> > ### Author Response · Authors · 2025-11-26
> >
> > > **How are "steps" and "attempts" formally defined? What operations occur per step, and what constitutes an attempt?**
> >
> > An inference-time learning step is one iteration of Algorithm 3's line 7 (*for each* step *do*). An attempted solution is the output of Algorithm 3, after ensembling the predictions that occur on line 16 of the Algorithm, which is not shown. This attempted solution is compared to a hidden ground truth to determine the score for that puzzle.
> >
> > > **How does the method compare to existing few-shot or zero-shot learning approaches on comparable tasks?**
> >
> > LLM-based solutions to ARC-AGI are generally zero-shot, since their learning of the test puzzle occurs via in-context learning instead of via gradient steps; methods using test-time training (TTT) can be considered few-shot approaches. Method-wise, these approaches use LLMs to solve the puzzle, do not rely on Occam's razor nor code golf, and generally bear no resemblance to our method. Performance-wise, they outperform CompressARC because they are pretrained on internet-scale data. We have not found any existing few/zero-shot approaches without prior training. We will provide these comparisons in the revised version of the paper.
> >
> > [1] Ronneberger, O., Fischer, P., & Brox, T. (2015, October). U-net: Convolutional networks for biomedical image segmentation. In International Conference on Medical image computing and computer-assisted intervention (pp. 234-241). Cham: Springer international publishing.

---

### Official Review · Reviewer_EfCx · 2025-11-02

**Soundness:** 2
**Presentation:** 3
**Contribution:** 2
**Rating:** 4
**Confidence:** 3

**Summary:**

This paper proposes CompressARC, a 76K-parameter neural model that tackles the ARC-AGI-1 benchmark without any pretraining or usage of the provided training set. The model applies the Minimum Description Length (MDL) principle to infer solutions purely at inference time, treating ARC puzzles as a code-golfing problem—that is, searching for the shortest program capable of reproducing the dataset. Despite its simplicity, CompressARC achieves 20% accuracy on the evaluation set and 34.75% on the training set, which is remarkable given that no pretraining or external data are used.

**Strengths:**

1. The framing of ARC-AGI solving as a code-golfing / MDL minimization problem is deeply original.
2. The implementation of inference-time learning via MDL provides a clean and theoretically motivated path to “training-free” intelligence.

**Weaknesses:**

1. Although the 20% accuracy result is interesting, it remains far from state-of-the-art (50%+).
2. Each puzzle takes 20 minutes and 2000 steps of inference-time optimization, which raises scalability and practicality concerns.
3. The discussion contrasts CompressARC mainly with large pretrained models (LLMs) but does not include comparisons to smaller ARC solvers or neuro-symbolic baselines.

**Questions:**

1. Compare with baseline models (random, heuristic, small CNN/VAE) under identical inference-time constraints.
2. Explicitly mention the restricted scalability and the dependency on ARC’s small grid size; discuss whether CompressARC could generalize to larger, more open-ended domains.

---

> ### Author Response · Authors · 2025-11-26
>
> We thank the reviewer for reading our work and providing their constructive criticism, and have made changes in the revised version of the paper, which is now uploaded, as reflected in our rebuttal.
>
> > **Although the 20% accuracy result is interesting, it remains far from state-of-the-art (50%+).**
>
> We acknowledge that the solve rate is indeed poor compared to SOTA. We would like frame this paper as a solution to produce data-efficient intelligence, which we will make clearer in the revised version of the introduction. This choice of framing is to align with the original intended goals of the creators of the ARC-AGI dataset: to build a system that exhibits skill acquisition with minimal input.
>
> > **Each puzzle takes 20 minutes and 2000 steps of inference-time optimization, which raises scalability and practicality concerns.**
>
> The amount inference-time compute is small for ARC-AGI, when compared to the LLM-based solutions used nowadays. The SOTA, OpenAI o3, takes >$200 of compute per puzzle. [1] We will give more context on the compute requirement in the revised version.
>
> > **Explicitly mention the restricted scalability and the dependency on ARC’s small grid size; discuss whether CompressARC could generalize to larger, more open-ended domains.**
>
> We thank the reviewer for raising the question of scalability and grid-size dependence. We would like to clarify that CompressARC's computational cost scales linearly with the total number of pixels, which is fundamentally not possible to improve upon, and is also better than the quadratic scaling behavior required by transformer-based methods in the ARC literature. In this sense, CompressARC is no more dependent on small ARC grids than existing approaches such as convolutional or transformer-based baselines, which also operate directly over the grid representation.
>
> Regarding runtime, the steps-per-second numbers in the paper may give the impression of limited scalability. This is an artifact of the current implementation rather than a fundamental constraint of the method. CompressARC introduces several custom operations (See Appendices C and D), and the prototype implementation does not use optimized CUDA kernels. As with most neural architectures, moving these operations to fused CUDA kernels or JIT-compiled kernels would yield speedups. We note that no algorithmic barrier prevents such optimization.
>
> We have added text to the paper clarifying these points and emphasizing that the current implementation reflects engineering trade-offs rather than a structural limitation of the method, and to discuss bringing CompressARC's advantages to other domains.
>
>
> > **The discussion contrasts CompressARC mainly with large pretrained models (LLMs) but does not include comparisons to smaller ARC solvers or neuro-symbolic baselines.**
>
> > **Compare with baseline models (random, heuristic, small CNN/VAE) under identical inference-time constraints.**
>
> We have included comparisons to a U-Net [2] that tries to few-shot learn in the same restrictive inference-time-only setting, and other works in the literature including brute force DSL searches [3], random, and LLM-based solutions [1]. These will be in Table 1 in the revised version.
>
> [1] Chollet, F. (2024, December 20). _OpenAI o3 Breakthrough High Score on ARC-AGI-Pub_. ARC Prize. https://arcprize.org/blog/oai-o3-pub-breakthrough
>
> [2] Ronneberger, O., Fischer, P., & Brox, T. (2015, October). U-net: Convolutional networks for biomedical image segmentation. In International Conference on Medical image computing and computer-assisted intervention (pp. 234-241). Cham: Springer international publishing.
>
> [3] Alijs. (2020). 5th place short notes [Kaggle discussion post]. Kaggle. https://www.kaggle.com/c/abstraction-and-reasoning-challenge/discussion/154377

---

### Official Review · Reviewer_Q3TN · 2025-11-04

**Soundness:** 2
**Presentation:** 1
**Contribution:** 2
**Rating:** 2
**Confidence:** 2

**Summary:**

The paper poses solving ARC-AGI problems as a search for a minimal program that reproduces the example outputs given the example inputs. The program found can then be applied to new inputs and its output be the solution to those new inputs. The benefit of this approach is it does not require one to train on a large dataset beforehand as training is done on a per-instance basis.

**Strengths:**

The justification for the approach using Occam’s razor is convincing. The ability to solve problems without pre-training is also appealing. Informative case studies are performed on success and failure cases.

**Weaknesses:**

The paper describes their approach in Algorithms 1, 2, and 3. There is a lot that is not clear in these algorithms.

-	These algorithms seek to minimize the seed length used. What is the relation between seed length and program complexity?

-	There are three seeds mentioned in Algorithm1 (seed_z1, seed_error, and seed_z2). Which seed length is the one that is to be minimized?

-	Algorithm 2 says “Measure n_exmpl,n_colors,width,height from P to initialize equivariant_NN”. What is meant by “measure”? How is this to be used to initialize the neural network? What is the initialization procedure?

-	How are \mu and \Sigma initialized?

The paper also does not contextualize the results in the broader research landscape. For someome unfamiliar with the ARC-AGI benchmark, one cannot know how these results compare to other existing results, especially for those that do not rely on pre-training. Is this method the only one that does not rely on pretraining?

**Questions:**

How does this approach compare to other methods that use pretraining and those that do not use pretraining?

See other questions in Weaknesses section.

---

> ### Author Response · Authors · 2025-11-26
>
> Thank you for your thoughtful review and for taking the time to evaluate our submission. We appreciate the opportunity to clarify an important misunderstanding and would like to clarify that the reviewer's summary does not fully reflect the mechanism we use. The minimal program does not reproduce the example outputs given the example inputs. Instead, the program is given no input, and it must produce both the example inputs and outputs, as well as the test inputs and outputs, by using internally hardcoded values. The program is not subsequently applied to the test input afterwards, as the entire puzzle, including both the demonstration and test inputs, is all outputted by the program at the same time. We have made changes to the new revision of the paper to clarify confusion as indicated by the questions, which we answer below:
>
> > **These algorithms seek to minimize the seed length used. What is the relation between seed length and program complexity?**
>
>
> The seed is hardcoded into the program, which means if the seed is longer, the program's code is longer, and is thus more complex.
>
>
> > **There are three seeds mentioned in Algorithm 1 (seed_z1, seed_error, and seed_z2). Which seed length is the one that is to be minimized?**
>
> The code in the algorithm is intended to repeat for all puzzles, so line 11 of Algorithm 1 illustrates the start of the next repetition, mirroring line 3 from before. This is why $seed_{z\,2}$ appears as a mirror of $seed_{z\,i}$. Since there are 400 puzzles, there will actually be 800 seeds (400 of $seed_{z\,i}$ and 400 of $seed_{error\,i}$). Since the program length is to be minimized, the sum of the lengths of all 800 seeds should be minimized, hence for every puzzle i, we are trying to minimize $len(seed_{z\,i}) + len(seed_{error\,i})$.
>
> > **Algorithm 2 says “Measure n_exmpl,n_colors,width,height from P to initialize equivariant_NN”. What is meant by “measure”? How is this to be used to initialize the neural network? What is the initialization procedure?**
>
> When we say "measure", we mean when the test puzzle P is given, we observe the properties of that puzzle. n_exmpl is the total number of demonstration + test input/output pairs, n_colors is the total number of unique colors in the puzzle (aside from black), width and height are the maximum width and height of any one grid in the given puzzle. We will change the wording to "Observe the dimensions ... of puzzle P".
>
> Indeed these measurements are _not_ used to initialize the neural network; they only affect the shapes of the learned $\mu, \Sigma$. Thank you for catching that typo, we will correct this in the revised version. The parameters in the network basically consist exclusively of linear layers, initialized with Xavier normal initialization. We will also add this detail in the revised version.
>
> > **How are $\mu$ and $\Sigma$ initialized?**
>
> In our actual method, $\mu$ and $\Sigma$ are defined through a reparameterization of two other variables which we define in Appendix C.1, which we use for stable training benefits. With this in consideration, we can essentially consider $\mu$ and $\Sigma$ as effectively initialized to:
>
> $$\mu \sim N(0, (1-\sigma^2) I), \quad \Sigma = \sigma^2 I, \quad \sigma = e^{-10^{4}/D}$$
>
> where $D$ is the number of elements in the initialized tensor. This initialization is picked to ensure $z \sim N(0,I)$, but also to make $KL(N(\mu, \Sigma)||N(0,I))=10^4$ at initialization, so that the loss component's contribution is not too large and yet has not collapsed. There are other details which we leave out here for clarity but which are described in the paper and supplementary code.

---

> > ### Author Response · Authors · 2025-11-26
> >
> > > **How does this approach compare to other methods that use pretraining and those that do not use pretraining?**
> >
> > Our method is the only one which uses a neural network with no pretraining, with nontrivial solve rates. LLMs, which use pretraining, have managed to achieve 87.5% on ARC-AGI, using astronomical amounts of inference-time compute through reasoning [1]. Other LLM-based techniques either use LLMs to directly output predicted grids in the form of tokenized grids [2, 3, 5], or have the LLMs generate code which manipulates the grids [2, 4].  This work often bears very little relationship to our work, with a few exceptions: (1) networks are often fine-tuned on individual test puzzles during test time, and (2) an ensembling procedure is used to produce the final prediction.
> >
> > Other methods that do not use pretraining typically rely on large-scale searches with hardcoded rule sets and domain-specific languages in order to find an analytical input/output rule that fits the demonstration examples [6]. These approaches share the commonality of heavily engineered hand-designed components that are made specifically to manipulate ARC-AGI puzzle data. They mostly do not rely on neural networks to solve the puzzle.
> >
> > We will make these comparisons more clear in the related work, which will also be moved to the top of the Appendix in the revised version.
> >
> > [1] Chollet, F. (2024, December 20). OpenAI o3 Breakthrough High Score on ARC-AGI-Pub. ARC Prize. https://arcprize.org/blog/oai-o3-pub-breakthrough
> >
> > [2] Li, W. D., Hu, K., Larsen, C., Wu, Y., Alford, S., Woo, C., ... & Ellis, K. (2024). Combining induction and transduction for abstract reasoning. arXiv preprint arXiv:2411.02272.
> >
> > [3] Barbadillo, G. (2024, December 8). Solution Summary – ARC24. Retrieved November 24, 2025, from https://ironbar.github.io/arc24/05_Solution_Summary/
> >
> > [4] Greenblatt, R. (2024, June 17). Getting 50% (SoTA) on ARC-AGI with GPT-4o. Redwood Research Blog. Retrieved November 24, 2025, from https://blog.redwoodresearch.org/p/getting-50-sota-on-arc-agi-with-gpt
> >
> > [5] Akyürek, E., Damani, M., Zweiger, A., Qiu, L., Guo, H., Pari, J., ... & Andreas, J. (2024). The surprising effectiveness of test-time training for few-shot learning. arXiv preprint arXiv:2411.07279.
> >
> > [6] Alijs. (2020). 5th place short notes [Kaggle discussion post]. Kaggle. https://www.kaggle.com/c/abstraction-and-reasoning-challenge/discussion/154377

---

### Author Response · Authors · 2025-11-26

**To all reviewers:**

We sincerely thank the reviewers for the time and effort spent evaluating our submission and for providing thoughtful and constructive feedback. We have carefully revised the paper in response to all comments. A new version has been uploaded in which the changes are highlighted in blue. Below, we summarize the main updates.

**Substantive revisions:**

- Expanded comparison to prior ARC-AGI methods in the Background section.
- Rewrote portions of the Introduction to give a more intuitive high-level description of our method.
- Added discussion on how learnings from CompressARC may transfer to other data-sparse domains.
- Expanded the theoretical foundation underlying the seed manipulation procedure; included additional references.
- Added baselines (with documentation).
- Reduced and consolidated some solution deconstruction figures (Figure 7) to save space.

**Clarity-focused revisions:**

- Clarified several algorithmic components and their roles.
- Improved the explanation in the Method section regarding why and how the dataset is code-golfed.
- Replaced the term “attempt” with “guess” where appropriate, and provided a precise definition of what constitutes a guess.
- Clarified the definition of a “step” in our system.
- Improved explanation of "measuring" the puzzle's dimensions and parameter initializations in the algorithms.
- Moved the previous Figure 1 into the Method section for better narrative flow.
- Made minor formatting and phrasing adjustments throughout for readability and consistency.

We hope these revisions adequately address the reviewers’ concerns and improve the clarity and strength of the manuscript. Thank you again for your constructive feedback.

---

### Meta-Review · Area_Chair_YrR3 · 2026-01-01

**Summary:**

This paper introduces a method: CompressARC, based on the minimum description length principle to solve ARC-AGI-1 problems. Notably, the proposed approach optimizes an equivariant neural network on individual puzzles, without requiring any large-scale training. Technically, the method overfits the neural network in a variational framework to the input–output pairs of a given problem, with the expectation that maximizing the likelihood of the outputs will yield the correct solution for the test pair, whose output is unknown. Experimental results on ARC-AGI problems show nearly 20% accuracy on test problems and approximately 35% accuracy on the training set.

**AC Comments:**
The paper received overall negative reviews, with two rejects, one borderline reject, and one borderline accept. While the reviewers consider the paper and its underlying idea to be timely—one reviewer (EfCx) even describing the use of MDL for ARC-AGI as “deeply original”—there are significant shortcomings in the exposition that make it unclear how the model actually produces solutions. Specifically, based on the technical details provided, the approach essentially trains a neural network to replicate the given input–output pairs. However, given the high-dimensional parameter space of neural networks and the very limited number of puzzle pairs available per ARC-AGI task, it is unclear why the model would learn the correct underlying rule, or what inductive bias in the setup guides the model toward a valid solution.

The authors are encouraged to revise the paper to address these issues and provide sufficient background and analysis to clarify the underlying mechanism. In addition, the paper should include proper comparisons to baseline methods, ablation studies (e.g., equivariant network versus transformer, varied network modules, confidence intervals on reported accuracy, network depth), and discussions of computational efficiency to better contextualize the benefits of the proposed approach. As noted by one reviewer (Y2bJ), it would also be valuable to explore whether the method generalizes to other problem settings, such as RAVEN progressive matrices. Furthermore, an analysis of the types of problems successfully solved by the approach (e.g., translations, color changes) would strengthen the contribution.

Overall, the paper requires a significant rewrite to clearly articulate and substantiate its contributions, and therefore cannot be accepted in its current form.

**Reviewer Concerns:**

*Reviewer Q3TN* raises several critical concerns regarding the proposed approach and missing technical details, including the lack of connections established between seed length and program complexity, unclear initialization of various algorithmic components, and insufficient contextualization of the results within the broader reasoning literature.

*Reviewer EfCx* points out that the reported performance of 20% is significantly below the state of the art, and notes that the paper should include baseline comparisons with neuro-symbolic approaches using comparable network sizes. The reviewer also questions the scalability of the method, given the time required to solve each problem.

*Reviewer Y2bJ* seeks further insight into whether the reported performance stems from genuine methodological innovation or careful engineering, particularly since the proposed method appears to rely on standard techniques. The reviewer also questions how generalization is achieved and highlights the lack of evaluation against state-of-the-art methods.

*Reviewer zdRF* notes the absence of theoretical insight into how the approach operates and questions whether it is specific to ARC-AGI or can generalize to other tasks. The reviewer also requests clarification on the efficiency of the method and comparisons to state-of-the-art approaches.

**Reviewer Scores:**

**Reviewer Q3TN:** Authors provide clarifications to their setup, including significant revisions to the submission. The authors also state that the proposed approach is the only one that does not use a neural network with any pre-training, and therefore argue that it is not directly comparable to prior methods.

[*AC's take on the response*]
While some technical questions and concerns regarding initialization are addressed, it is unlikely that the reviewer would have been fully satisfied by the responses, as the methodology still appears questionable and lacks insight into why it should work under such low-data regimes.

**Reviewer EfCx:** Authors argue that although the reported performance is significantly below the state of the art, the approach can be viewed as a form of data-efficient intelligence. They also clarify that the reported computational inefficiency stems from an unoptimized implementation and could be improved with better engineering choices. Additional comparisons to U-Net and non-learning-based alternatives such as brute-force and random search are included, suggesting that CompressARC offers a reasonable trade-off.

[*AC's take on the response*]
Despite some important revisions, it is unlikely that the reviewer would have raised the score, given the substantially below–state-of-the-art performance and the lack of compelling baselines that provide deeper insight into the reported results. The absence of ablation studies further limits the strength of the claims.

**Reviewer Y2bJ:** Authors provide additional results using U-Net, showing lower performance compared to CompressARC. They also offer clarifications and references to sections of the paper where the technical questions raised by the reviewer are addressed.

[*AC's take on the response*]
Given the lack of new insights introduced in the rebuttal, AC believes it is unlikely that the reviewer would have been satisfied by the authors’ responses.

**Reviewer zdRF:** Authors revise the paper to include additional references that contextualize the idea of relative entropy coding, noting that expected seed length can be reduced by optimizing the KL divergence between distributions. The authors also clarify aspects related to the efficiency of the approach. The reviewer notes that these concerns are only partially addressed and maintains the original score.

---

### Decision · Program_Chairs · 2026-01-26

Reject